# Distributional Alignment Games for Answer-Level Fine-Tuning

**Mehryar Mohri** [1 2]   **Jon Schneider** [1]   **Yifan Wu** [3]

## Abstract

We focus on the problem of *Answer-Level Fine-Tuning* (ALFT), where the goal is to optimize a language model based on the correctness or properties of its final answers, rather than the specific reasoning traces used to produce them. Directly optimizing answer-level objectives is computationally intractable due to the need to marginalize over the vast space of latent reasoning paths. To overcome this, we propose a general game-theoretical framework that lifts the problem to a *Distributional Alignment Game*. We formulate ALFT as a two-player game between a Policy (the generator) and a Target (an auxiliary distribution). We prove that the Nash Equilibrium of this game corresponds exactly to the solution of the original answer-level optimization problem. This variational perspective transforms the intractable marginalization problem into a tractable projection problem. We demonstrate that this framework unifies recent approaches to diversity and self-improvement (coherence) and provide efficient algorithms compatible with Group Relative Policy Optimization (GRPO), such as COHERENCE-GRPO, yielding significant complexity gains in mathematical reasoning tasks.

## 1. Introduction

In reasoning-intensive domains such as mathematics, code generation, and open-ended question answering, the primary utility of a Large Language Model (LLM) lies in the correctness of its final *answer* $z$, rather than the specific phrasing of the intermediate *reasoning trace* $y$. This distinction has driven a surge of interest in Chain-of-Thought (CoT) prompting (Wei et al., 2022) and reasoning-focused

alignment strategies. While "Process Supervision" aims to guide the model step-by-step (Uesato et al., 2022; Lightman et al., 2023), "Outcome Supervision" focuses purely on the final result, allowing the model the flexibility to discover diverse valid reasoning paths (Wang et al., 2023). This latter setting defines the problem of *Answer-Level Fine-Tuning (ALFT)*. An ideal reasoning policy should be robust yet flexible: free to explore various trajectories provided they converge to correct or consistent outcomes.

However, ALFT presents a fundamental computational bottleneck that is absent in standard trace-level fine-tuning. Standard alignment methods, such as Supervised Fine-Tuning (SFT) or Direct Preference Optimization (DPO) (Rafailov et al., 2023), optimize the likelihood of a specific, observed trace $y$. In contrast, ALFT requires optimizing the *marginal probability* of the answer $z$. Because the mapping from reasoning trace to answer $z = \mathsf{E}(y)$ is often non-differentiable (e.g., executing a program) and many-to-one, computing the gradient of the marginal likelihood requires marginalizing over the vast combinatorial space of all latent traces consistent with $z$. This makes direct optimization intractable and standard gradient estimators (like REINFORCE) prohibitively high-variance, often requiring sophisticated variance reduction techniques or prohibitively large sample sizes to stabilize training.

To overcome this, we propose a fundamental shift in perspective. Instead of attacking the marginalization sum directly, we lift the optimization problem to a *Distributional Alignment Game*. We introduce an auxiliary *Target Distribution* q, which serves as a variational proxy for the intractable marginals. By leveraging Fenchel duality, we formulate ALFT as a two-player game between a *Policy* $\pi$ (the generator) and a *Target* q. The Policy attempts to minimize the divergence of its trace distribution from the Target, while the Target adapts to satisfy high-level distributional properties—such as enforcing consensus (coherence), maximizing coverage (diversity), or satisfying safety constraints—thereby guiding the Policy. This variational perspective transforms the hard marginalization problem into a tractable projection problem, where the difficulty is offloaded to the *Target Step*: finding the optimal q that best challenges or guides the current policy.

Crucially, this framework provides a unified theoretical lens

---
[1]Google Research, New York, NY; [2]Courant Institute of Mathematical Sciences, New York, NY [3]Microsoft Research. Correspondence to: Mehryar Mohri <mohri@google.com>, Jon Schneider <jschnei@google.com>, Yifan Wu <yifan.wu@u.northwestern.edu>.

*Proceedings of the $43^{rd}$ International Conference on Machine Learning*, Seoul, South Korea. PMLR 306, 2026. Copyright 2026 by the author(s).

for understanding recent heuristic successes in reasoning alignment. We demonstrate that distinct alignment goals are merely different instantiations of the game's Target Step. For instance, we show that the *diversity-promoting* objective of maximizing entropy corresponds to a game against an adversarial target, where the Nash Equilibrium justifies the "Inverse-Frequency Reward" heuristic recently proposed by Li et al. (2025). Conversely, the *coherence-promoting* objective (Self-Improvement) corresponds to a cooperative game, where the equilibrium is the Bregman Centroid of the generated distributions. This rigor allows us to replace ad-hoc reward engineering with derived solutions to well-posed optimization problems.

We further bridge the gap between theory and practice by deriving a scalable algorithmic framework. We show that the policy update steps in our game are compatible with Group Relative Policy Optimization (GRPO) (Shao et al., 2024), an efficient policy gradient method that operates on groups of outputs. By deriving rewards directly from the optimal Target distribution $q^*$, we instantiate algorithms like COHERENCE-GRPO that solve the alignment game without explicit marginalization.

Our main contributions are as follows:

1. **General Framework:** We formalize ALFT as a min-max game that is convex-concave in the dual variables. We prove *consistency*: the Nash Equilibrium of this game recovers the exact optimal solution to the original answer-level problem, identifying the optimal policy as a projection of the reference onto the target-compatible traces.

2. **Unification of Objectives:** We provide a single mathematical umbrella for disparate alignment goals. We prove that maximizing diversity implies an inverse-frequency update, while minimizing incoherence implies a consensus-based update, grounding recent empirical methods in convex duality theory.

3. **Scalable Algorithms:** We propose game-theoretic variants of GRPO. We introduce COHERENCE-GRPO for discrete domains and PAIRWISE-GRPO for open-ended domains, demonstrating how to compute rigorous advantage signals from group statistics.

4. **Extensions to Safety and Continuous Domains:** We extend the framework beyond standard exact matching. We introduce a pairwise formulation for open-ended generation where exact answers are undefined, and a Primal-Dual algorithm for enforcing distributional constraints (e.g., safety and fairness) by interpreting Lagrange multipliers as dynamic penalty weights.

5. **Empirical Validation:** We conduct experiments on self-improvement of language models entirely without ground truth, testing COHERENCE-GRPO and PAIRWISE-GRPO with three 3B-class instruction-tuned LLMs on GSM8K and TriviaQA. On GSM8K, accuracy improves by +3.18 to +9.18 percentage points; on TriviaQA, PAIRWISE-GRPO yields up to +42.06% relative EM. We also report a pilot validation of SAFETY-GRPO, demonstrating that the primal-dual update successfully enforces safety constraints.

## 2. General Framework: Alignment as a Distributional Game

### 2.1. Problem: Answer-Level Fine-Tuning

Let $\mathcal{X}$ be the space of input prompts, $\mathcal{Y}$ the space of reasoning traces, and $\mathcal{Z}$ the space of final answers. A deterministic *extraction function* $\mathsf{E}: \mathcal{Y} \to \mathcal{Z}$ maps traces to answers. A policy $\pi(y|x)$ (a conditional distribution over traces) induces a marginal distribution over answers:

$$\nu_\pi(z|x) = \sum_{y \in \mathsf{E}^{-1}(z)} \pi(y|x).$$

Our goal is to find a policy $\pi$ that minimizes a loss $\mathcal{L}$ that depends only on this answer distribution $\nu_\pi$, subject to a regularization constraint that keeps the policy close to a reference $\pi_0$. We formulate this as the following convex optimization problem over a convex set $\Pi \subseteq \Delta(\mathcal{Y})^{\mathcal{X}}$:

$$\min_{\pi \in \Pi} \mathcal{J}(\pi) = \mathbb{E}_x\big[\mathcal{R}(\nu_\pi(\cdot|x)) + \beta\, \mathsf{D}_{\mathrm{KL}}(\pi(\cdot|x)\|\pi_0(\cdot|x))\big], \text{(1)}$$

where the functional $\mathcal{R}: \Delta(\mathcal{Z}) \to \mathbb{R}$ encodes our answer-level goal (e.g., minimizing entropy for coherence, or maximizing entropy for diversity) and where $\beta > 0$ is a hyperparameter. We assume $\mathcal{R}$ is convex and lower semi-continuous (l.s.c.) to ensure the existence of solutions and applicability of duality. The overall objective $\mathcal{J}(\pi)$ is strictly convex due to the KL term, guaranteeing a unique optimal policy.

**Why is this hard?** Gradient-based optimization of (1) is challenging because the objective $\mathcal{R}$ acts on the marginal $\nu_\pi$. Computing the gradient requires backpropagating through the summation $\nu_\pi(z) = \sum_{y \in \mathsf{E}^{-1}(z)} \pi(y)$. Since the set $\mathsf{E}^{-1}(z)$ is unknown and vast, we cannot compute this sum or its gradient efficiently. Monte-Carlo estimates suffer from extreme variance because the "credit" for a good answer $z$ must be assigned to the specific trace $y$ that produced it, without knowing if other traces would have done better.

### 2.2. Equivalent Formulation as Min-Max Game via Fenchel Duality

We now show how to transform the intractable marginalization problem into a tractable game. We rely on the Fenchel Duality Theorem (Rockafellar, 1997).

Let $\mathcal{R}^*: \mathbb{R}^{|\mathcal{Z}|} \to \mathbb{R}$ denote the Fenchel conjugate of $\mathcal{R}$, defined as:

$$\mathcal{R}^*(u) = \sup_{\nu \in \Delta(\mathcal{Z})} \{\langle \nu, u \rangle - \mathcal{R}(\nu)\}.$$

Since $\mathcal{R}$ is convex and lower semi-continuous, the Fenchel-Moreau theorem (Rockafellar, 1997) guarantees that $\mathcal{R}$ equals its biconjugate $\mathcal{R}^{**}$:

$$\mathcal{R}(\nu) = \sup_{u \in \mathbb{R}^{|\mathcal{Z}|}} \{\langle \nu, u \rangle - \mathcal{R}^*(u)\}. \tag{2}$$

This variational form allows us to introduce an auxiliary variable $u$ to decouple the marginalization. Substituting (2) into the primal objective (1):

$$\min_{\pi \in \Pi} \mathbb{E}_{x \in \mathbb{R}^{|\mathcal{Z}|}} \left[ \sup_u (\langle \nu_\pi(\cdot|x), u \rangle - \mathcal{R}^*(u)) \right.$$
$$\left. + \beta \, \mathsf{D}_{\mathrm{KL}}(\pi(\cdot|x) \parallel \pi_0(\cdot|x)) \right].$$

The game is naturally defined over the dual variables $u \in \mathbb{R}^{|\mathcal{Z}|}$. However, the objective is invariant under constant shifts. First, observe that for any functional $\mathcal{R}$ defined on the probability simplex $\Delta(\mathcal{Z})$, the Fenchel conjugate satisfies the shift property $\mathcal{R}^*(u + c\mathbf{1}) = \mathcal{R}^*(u) + c$. This holds because $\nu \in \Delta(\mathcal{Z})$ implies $\sum \nu(z) = 1$, so:

$$\mathcal{R}^*(u + c\mathbf{1}) = \sup_{\nu \in \Delta(\mathcal{Z})} \{\langle \nu, u + c\mathbf{1} \rangle - \mathcal{R}(\nu)\}$$
$$= \sup_{\nu \in \Delta(\mathcal{Z})} \left\{ \langle \nu, u \rangle - \mathcal{R}(\nu) + c \sum_z \nu(z) \right\}$$
$$= \mathcal{R}^*(u) + c.$$

Thus, the dual objective term $\langle \nu_\pi, u \rangle - \mathcal{R}^*(u)$ is invariant under the transformation $u \leftarrow u + c\mathbf{1}$, as the linear shift $c$ cancels exactly with the conjugate shift. This redundancy implies that the effective dual space is the quotient space $\mathbb{R}^{|\mathcal{Z}|}/\mathrm{span}(\mathbf{1})$. We can therefore parameterize the dual variables without loss of generality by choosing a canonical representative from each equivalence class. Specifically, we use the parameterization $u(z) = -\beta \log \mathsf{q}(z)$ with $\mathsf{q} \in \mathrm{int}(\Delta(\mathcal{Z}))$. This mapping establishes a bijection between the quotient space and the interior of the simplex. For any arbitrary vector $u \in \mathbb{R}^{|\mathcal{Z}|}$, there exists a unique scalar shift $c$ that satisfies the probability normalization constraint $\sum_z \exp(-(u(z) + c)/\beta) = 1$. Solving for $c$ reveals that this unique shift is determined exactly by the Log-Sum-Exp function $c = \beta \log(\sum_{z \in \mathcal{Z}} \exp(-u(z)/\beta))$.

By setting the canonical representative to $v = u + c\mathbf{1}$, we guarantee that $\mathsf{q}(z) = \exp(-v(z)/\beta)$ is a valid normalized distribution. We define the transformed conjugate functional

$\Psi(\mathsf{q}) = \mathcal{R}^*(-\beta \log \mathsf{q})$. The coupling term becomes:

$$\langle \nu_\pi(\cdot|x), -\beta \log \mathsf{q} \rangle = -\beta \sum_{z \in \mathcal{Z}} \nu_\pi(z|x) \log \mathsf{q}(z)$$
$$= -\beta \sum_{z \in \mathcal{Z}} \left[ \sum_{y \in \mathsf{E}^{-1}(z)} \pi(y|x) \right] \log \mathsf{q}(z)$$
$$= -\beta \sum_{z \in \mathcal{Z}} \sum_{y \in \mathsf{E}^{-1}(z)} \pi(y|x) \log \mathsf{q}(\mathsf{E}(y))$$
$$\text{(Since } \mathsf{E}(y) = z \text{ for } y \in \mathsf{E}^{-1}(z))$$
$$= -\beta \sum_{y \in \mathcal{Y}} \pi(y|x) \log \mathsf{q}(\mathsf{E}(y))$$
$$\text{(Partition property: } \mathcal{Y} = \bigcup_z \mathsf{E}^{-1}(z))$$
$$= -\beta \mathbb{E}_{y \sim \pi(\cdot|x)} [\log \mathsf{q}(\mathsf{E}(y))].$$

Substituting this back, we obtain the following min-max objective function $\mathcal{G}(\pi, \mathsf{q})$:

$$\mathcal{G}(\pi, \mathsf{q}) \triangleq \mathbb{E}_x \left[ \beta \mathsf{D}_{\mathrm{KL}}(\pi(\cdot|x) \parallel \pi_0(\cdot|x)) \right.$$
$$\left. - \beta \mathbb{E}_{y \sim \pi(\cdot|x)} [\log \mathsf{q}(\mathsf{E}(y))] - \Psi(\mathsf{q}) \right]. \tag{3}$$

### 2.3. Theoretical Consistency

The derivation above leads directly to our main result: the equilibrium of the game defined by $\mathcal{G}$ recovers the solution to the original hard answer-level problem.

**Theorem 2.1** (Consistency of the Game). *Let $\mathcal{R}: \Delta(\mathcal{Z}) \to \mathbb{R}$ be a convex, lower semi-continuous functional. Then:*

1. *The primal problem (1) is equivalent to the min-max game: $\min_{\pi \in \Pi} \mathcal{J}(\pi) = \min_{\pi \in \Pi} \max_{\mathsf{q} \in \Delta(\mathcal{Z})} \mathcal{G}(\pi, \mathsf{q})$.*

2. *For a fixed Target distribution $\mathsf{q}$, the optimal policy $\pi^*$ is the unique projection of $\pi_0$ onto the traces compatible with $\mathsf{q}$: $\pi^*(y|x) \propto \pi_0(y|x)\mathsf{q}(\mathsf{E}(y)|x)$.*

*Proof. Equivalence:* This follows directly from the Fenchel Duality Theorem. Since $\mathcal{R}$ is convex and l.s.c., $\mathcal{R}(\nu_\pi) = \sup_{\mathsf{q}}(-\beta\langle \nu_\pi, \log \mathsf{q} \rangle - \Psi(\mathsf{q}))$. Substituting this into the definition of $\mathcal{J}(\pi)$ yields $\min_\pi \max_{\mathsf{q}} \mathcal{G}(\pi, \mathsf{q})$.

*Optimal Policy Form:* Consider minimizing $\mathcal{G}(\pi, \mathsf{q})$ with respect to $\pi$ for a fixed $\mathsf{q}$. The relevant terms are:

$$\min_{\pi \in \Pi} \mathbb{E}_x \left[ \beta \mathbb{E}_{y \sim \pi} \left[ \log \frac{\pi(y)}{\pi_0(y)} \right] - \beta \mathbb{E}_{y \sim \pi} [\log \mathsf{q}(\mathsf{E}(y))] \right]$$
$$= \min_{\pi \in \Pi} \beta \mathbb{E}_x \left[ \mathbb{E}_{y \sim \pi} \left[ \log \frac{\pi(y)}{\pi_0(y)\mathsf{q}(\mathsf{E}(y))} \right] \right].$$

This is equivalent to minimizing $\mathsf{D}_{\mathrm{KL}}(\pi \parallel \tilde{\mathsf{q}})$, where $\tilde{\mathsf{q}}(y) \propto \pi_0(y)\mathsf{q}(\mathsf{E}(y))$. $\qquad\square$

We further show that finding an approximate equilibrium of the game yields a good solution to the original problem.

**Proposition 2.2** (Approximation Guarantee). *Let* $(\widehat{\pi}, \widehat{q})$ *be an $\epsilon$-approximate equilibrium of the game $\mathcal{G}$, such that $\mathcal{G}(\widehat{\pi}, \widehat{q}) \leq \min_\pi \max_q \mathcal{G}(\pi, q) + \epsilon$. Then $\widehat{\pi}$ is an $\epsilon$-approximate minimizer of the primal objective $\mathcal{J}(\pi)$.*

*Proof.* By Part 1 of the Theorem, $\mathcal{J}(\pi) = \max_q \mathcal{G}(\pi, q)$. If $(\widehat{\pi}, \widehat{q})$ is an $\epsilon$-equilibrium, then $\mathcal{J}(\widehat{\pi}) \leq \mathcal{J}(\pi^*) + \epsilon$. $\qquad\square$

Finding an equilibrium of this minimax game is far more tractable than the original primal problem. Crucially, the objective $\mathcal{G}(\pi, q)$ is strictly convex in $\pi$ (due to the KL term) and concave in the natural dual parameters. This strict convex-concave structure ensures that Alternating Best Response converges to the unique global optimum $(\pi^*, q^*)$ (Tseng, 2001), and that general no-regret dynamics converge at rate $O(1/\sqrt{T})$; see Appendix E for details.

### 2.4. Specific Instantiations

The power of our framework lies in its universality. By selecting the functional $\mathcal{R}$, we recover distinct alignment paradigms as specific instantiations of the game's Target Step:

**Standard RL.** When the goal is simply to maximize a scalar reward $r(z)$ (e.g., $r(z) = \mathbb{I}[z = z_{\text{gold}}]$), the answer-level functional is linear: $\mathcal{R}(\nu) = -\mathbb{E}_{z \sim \nu}[r(z)]$. In this case, the convex conjugate $\mathcal{R}^*(u)$ collapses the dual space to a *single point*, fixing the optimal Target distribution to the exponentiated reward distribution: $q^*(z) \propto \exp(r(z)/\beta)$ (see Proposition 3.1).

**Supervised Fine-Tuning.** One interpretation of SFT is that we want to train the model to match a specific ground truth Dirac distribution $\delta_{\text{ground truth}}$. This is captured by the functional $\mathcal{R}(\nu) = D_{\text{KL}}(\delta_{\text{ground truth}} \parallel \nu)$. The corresponding game is trivial; the Target is fixed to the ground truth $q^*(z) = \delta_{\text{ground truth}}(z)$.

**Diversity.** When the goal is exploration, we choose $\mathcal{R}(\nu) = -H(\nu)$ (negative entropy). The Target q acts as an *adversary*, maximizing the dual potential $\Psi(q)$ (the log-partition function). The optimal strategy $q^*$ assigns high mass to under-represented answers, forcing the policy to spread its probability mass. As we show in Appendix C, this recovers the "Inverse-Frequency" heuristic (Li et al., 2025).

**Safety.** The goal of safety can be cast as requiring that the policy $\pi$ lies in a "safe" set $\mathcal{C}$. This can be enforced by the functional $\mathcal{R}(\nu) = \text{Ind}_\mathcal{C}(\nu)$ (taking value 0 if $\nu \in \mathcal{C}$ and $+\infty$ otherwise). For this $\mathcal{R}$, the corresponding target $q^* = \text{argmin}_{q \in \mathcal{C}} D_{\text{KL}}(q \parallel \nu_\pi)$, the information projection of q onto $\mathcal{C}$. See Appendix D for details.

**Coherence.** Finally, we remark briefly here that (with minor changes to the setup) the goal of self-improvement via consensus (Mohri et al., 2025) can be captured by setting $\mathcal{R}$ to a specific expected divergence. We discuss this in greater detail in Section 4.

## 3. Generic Algorithm: Solving the Game via GRPO

We now describe practical algorithms for solving the Distributional Alignment Game. We use *Group Relative Policy Optimization (GRPO)* as the solver for the Policy Step, feeding it rewards derived from the Target Step.

*Group Relative Policy Optimization (GRPO)* is a policy gradient algorithm designed for scenarios where evaluation occurs over a *group* of outputs. For each input $x$, GRPO samples a group of $K$ outputs $G = \{y_1, \ldots, y_K\}$ generated by the current policy $\pi_{\text{old}}$. It updates the policy using a surrogate objective that relies on advantages $A_i$ computed relative to the group average:

$$\mathcal{L}_{\text{GRPO}}(\pi)$$
$$= \mathbb{E}_x \left[ \mathbb{E}_{G \sim \pi_{\text{old}}} \left[ \frac{1}{K} \sum_{i=1}^{K} \frac{\pi(y_i|x)}{\pi_{\text{old}}(y_i|x)} A_i - \beta D_{\text{KL}}(\pi \| \pi_{\text{ref}}) \right] \right]. \quad (4)$$

Crucially, GRPO is an optimization engine that requires an external definition of *advantage* or *reward*. In our framework, we define a general algorithm by deriving these advantages directly from the game's equilibrium condition.

Recall that the optimal ALFT policy satisfies $\pi^*(y|x) \propto \pi_0(y|x) q^*(E(y)|x)$ (Theorem 2.1). This suggests that the raw *reward* for a trace $y$ should be determined by the likelihood of its answer under the optimal Target distribution $q^*$. We therefore define the reward for the $i$-th trace in the group as:

$$R_i = \log q^*(E(y_i)|x).$$

The advantage $A_i$ is then defined as the standardized reward relative to the group statistics, which serves as a baseline to reduce variance:

$$A_i = \frac{R_i - \text{Mean}(\{R_1, \ldots, R_K\})}{\text{StdDev}(\{R_1, \ldots, R_K\}) + \epsilon}, \quad (5)$$

where $\epsilon > 0$ is a small positive number. An advantage $A_i > 0$ indicates that trace $y_i$ aligns better with the Target than the average trace in the group. This general algorithm can be applied to any distributional alignment problem where the optimal Target $q^*$ can be estimated. We now present two specific instantiations of this algorithm.

### 3.1. Theoretical Justification for GRPO

To understand why GRPO solves the game, consider the **Policy Step** derived in Theorem 2. For a fixed target $q^*$,

the optimal policy $\pi^*$ must minimize the divergence to the lifted target $\tilde{q}(y|x) \propto \pi_0(y|x)q^*(E(y)|x)$. This minimization problem, $\arg\min_\pi D_{KL}(\pi\|\tilde{q})$, can be rewritten as maximizing an expected reward subject to a KL constraint:

$$\arg\min_\pi \mathbb{E}_{y\sim\pi}\left[\log\frac{\pi(y)}{\pi_0(y)q^*(E(y))}\right]$$
$$= \arg\max_\pi \mathbb{E}_{y\sim\pi}\left[\underbrace{\log q^*(E(y))}_{\text{Reward }R(y)} - \underbrace{\log\frac{\pi(y)}{\pi_0(y)}}_{\text{KL Penalty}}\right],$$

an Entropy-Regularized Reinforcement Learning problem where the reward function is defined by the Target distribution:

$$R(y) = \beta\log q^*(E(y)). \tag{6}$$

GRPO is an efficient gradient estimator for exactly this type of objective. It estimates the policy gradient using a group of samples $G = \{y_1, \ldots, y_K\}$ and a group-based baseline to reduce variance. Crucially, the Advantage formulation in GRPO (Eq. 5) can be viewed as an empirical estimate of the gradient direction derived from the Nash Equilibrium condition.

This connection to entropy-regularized RL suggests a deeper relationship with recent advancements in preference optimization. We formalize this link in the following proposition, which establishes our framework as the dual of Direct Preference Optimization (DPO).

**Proposition 3.1** (Duality with Direct Preference Optimization)**.** *The Distributional Alignment Game is the mathematical dual of Direct Preference Optimization (DPO) for answer-level objectives. Specifically, the optimal target distribution $q^*$ corresponds to the exponentiated ground-truth reward function $r^*$ of the equivalent RL problem:*

$$q^*(E(y)) = \frac{1}{Z}\exp\left(\frac{r^*(y)}{\beta}\right), \tag{7}$$

*where $Z$ is the partition function. Thus, while DPO eliminates the reward model to solve for the policy directly, our framework eliminates the policy to solve for the optimal target distribution.*

*Proof.* In the standard RL setting used by DPO (Rafailov et al., 2023), the optimal policy $\pi^*$ maximizing a reward $r^*$ subject to KL-regularization from $\pi_0$ has the closed form:

$$\pi^*(y) \propto \pi_0(y)\exp\left(\frac{r^*(y)}{\beta}\right).$$

In our game-theoretic framework (Theorem 1), we derived that for a fixed optimal target $q^*$, the optimal policy satisfies:

$$\pi^*(y) \propto \pi_0(y)q^*(E(y)).$$

Equating these two forms immediately yields the relation $q^*(E(y)) \propto \exp(r^*(y)/\beta)$. This proves that learning the optimal target distribution $q^*$ is equivalent to learning the implicit reward function of the underlying decision process. $\square$

**Implication for DPO.** This duality suggests that answer-level objectives could theoretically be solved using DPO. However, unlike standard alignment where preferences are static, answer-level objectives like diversity or coherence imply a *dynamic* reward function that depends on the current policy's marginals (e.g., the inverse-frequency reward changes as the model changes). Therefore, a DPO-based solution would require an *iterative* formulation—similar to Self-Play DPO, where preference pairs are continuously re-generated and re-labeled based on the current optimal target $q^*$. Our GRPO-based algorithm can be viewed as a more direct online solver for this iterative game, bypassing the explicit construction of preference datasets.

### 3.2. Generic Algorithm: GAME-GRPO

Based on this justification, we define a generic algorithm, GAME-GRPO. At each step, we first compute the optimal Target $q^*$ (Target Step), which defines the reward $R$. We then compute advantages and update the policy (Policy Step).

The Advantage $A_i$ for a trace $y_i$ is defined as its standardized reward relative to the group statistics:

$$A_i = \frac{R(y_i) - \text{Mean}(\{R(y_j)\}_{j=1}^K)}{\text{StdDev}(\{R(y_j)\}_{j=1}^K) + \epsilon}. \tag{8}$$

The policy update maximizes the surrogate objective:

$$\mathcal{L}_{\text{GRPO}}(\pi)$$
$$= \mathbb{E}_x\left[\mathbb{E}_{G\sim\pi_{\text{old}}}\left[\frac{1}{K}\sum_{i=1}^K\frac{\pi(y_i|x)}{\pi_{\text{old}}(y_i|x)}A_i - \beta D_{KL}(\pi\|\pi_{\text{ref}})\right]\right].$$

Thus, our general algorithmic framework consists of alternating two steps (illustrated in Figure 1):

1. **Target Step:** Estimate the optimal $q^*$ from the sampled group $G$ (e.g., via majority vote or centrality).

2. **Policy Step:** Calculate rewards $R_i = \log q^*(E(y_i))$ and perform a GRPO update.

## 4. Answer-Level Self-Improvement via Coherence

### 4.1. Background

We briefly recall the coherence framework for self-improvement; see Appendix G for a full treatment.

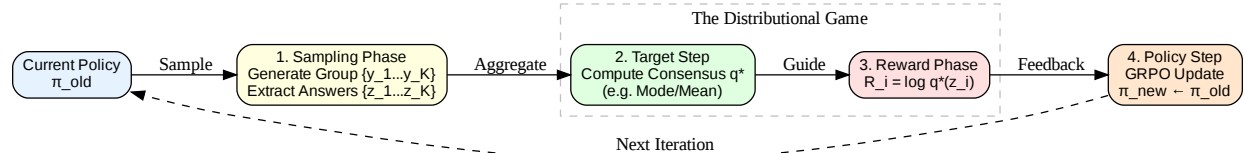

*Figure 1.* The Game-Theoretic Alignment Loop. This diagram illustrates the alternating best-response dynamics used in our GRPO-based algorithms. (1) The *Policy* generates a group of traces. (2) The *Target Step* solves the game by aggregating these outputs into an optimal target distribution q* (e.g., via consensus for coherence or inverse-frequency for diversity). (3) This target defines the reward signal. (4) The *Policy Step* updates the model to align with the target.

Standard self-improvement methods typically rely on heuristic filtering or scalar rewards. In contrast, the coherence framework proposed by Mohri et al. (2025) offers a rigorous geometric approach grounded in the principle of *coherence*. The core idea is that a reliable model should produce mutually consistent distributions for inputs that are semantically equivalent (e.g., a prompt $x$ and its paraphrase $\Phi(x)$). Formally, let $\Phi : \mathcal{X} \to \mathcal{X}$ be a task-preserving transformation[1]. The set of *coherent policies* is defined as $\mathcal{C}_{\text{coh}} = \{\pi \mid \nu_\pi(\cdot|x) = \nu_\pi(\cdot|\Phi(x))\}$.

Importantly, as with the other answer-level tasks in this paper, it is the induced distribution $\nu_\pi$ on answers that we require to be coherent, as opposed to the immediate distribution $\pi(\cdot|x)$ on sequences $y$. In particular, for a task like a math problem, $y$ represents the chain-of-thought used to solve the problem and $z$ the numerical result. Our training algorithm should minimize the incoherence of the answer distributions on two equivalent problems $x$ and $\phi(x)$ while maintaining the diversity of valid reasoning paths in $\mathcal{Y}$.

Self-improvement is cast as a constrained optimization problem: finding the coherent policy $\widehat{\pi}$ that is closest to the initial baseline $\pi_0$ measured by a Bregman divergence $D_F$ (such as the KL divergence):

$$\widehat{\pi} = \underset{\pi \in \mathcal{C}_{\text{coh}}}{\arg\min} \, \mathbb{E}_x[D_F(\pi(\cdot|x) \parallel \pi_0(\cdot|x))].$$

This operation is a *Bregman Projection*. Mohri et al. (2025) prove that this projection guarantees *monotonic improvement*: the new policy $\widehat{\pi}$ is strictly closer to the optimal ground-truth policy $\pi^*$ than the baseline $\pi_0$ was, provided $\pi^*$ is itself coherent.

This minimization problem can also be captured as a slight generalization of the ALFT framework, where we allow the functional $\mathcal{R}$ (previously a functional acting on distributions over $\mathcal{Z}$ ($\nu_\pi(\cdot|x)$)) to act on the collection of distributions in an equivalence class. In particular, if we define $\mathcal{R}(\{\nu_\pi(\cdot|x')\}_{x' \in \mathcal{O}_x})$ to equal 0 if all distributions $\nu_\pi(\cdot|x')$ are equal and $+\infty$ otherwise (i.e., $\mathcal{R}$ is the indicator function of the restriction of $\mathcal{C}_{\text{coh}}$ to the orbit of $x$),

---

[1]Following (Mohri et al., 2025), we assume for mathematical simplicity that $\Phi$ is an involution, i.e., that $\Phi(\Phi(x)) = x$.

then the above minimization problem agrees with the minimax problem defined in Theorem 2.1. The corresponding consensus target q* is given by the geometric mean $q^*(z) = \frac{1}{Z}\left(\prod_{x' \in \mathcal{O}_x} \nu(z|x')\right)^{\frac{1}{|\mathcal{O}_x|}}$, where $Z$ is a normalizing factor.

### 4.2. Theoretical Analysis of Consensus Targets

The core of the Coherence Game lies in defining the optimal Target q* that the policy must match. While the *Geometric Mean* is the exact solution for minimizing KL divergence, it is computationally intractable for large output spaces due to the partition function $Z$ (Mohri et al., 2025). To derive a practical algorithm, we analyze two tractable approximations:

**1. The Arithmetic Mean.** We can relax the objective to minimize the squared Euclidean distance, yielding the Arithmetic Mean $q_{\text{AM}}(z) = \frac{1}{|\mathcal{O}_x|}\sum_{x' \in \mathcal{O}_x} \nu(z|x')$. Crucially, we can prove that this relaxation remains theoretically sound.

Let $K = |\mathcal{O}_x|$ and let $\{\nu_1, \nu_2, \ldots, \nu_K\}$ be an enumeration of the distributions $\nu(z|x')$ for $x' \in \mathcal{O}_x$. Let $H^2(\{\nu_k(z)\}) = 1 - \sum_{z \in \mathcal{Z}}\left(\prod_{k=1}^{K}\nu_k(z)\right)^{1/K}$ denote the Generalized Squared Hellinger (GSH) distance, which is a measure of the diversity of the distributions $\nu_k$. Let the improvement of a solution $\pi$ with respect to the baseline $\pi_0$ be defined as

$$\text{Improv}(\pi) = \mathbb{E}[D_{\text{KL}}(\pi^*(x) \parallel \pi_0(x))] \\ - \mathbb{E}[D_{\text{KL}}(\pi^*(x) \parallel \pi(x))].$$

Then, the following guarantee holds.

**Theorem 4.1** (Stability of Arithmetic Consensus)**.** *Let $\widehat{\pi}_{Hybrid}$ be the policy resulting from projection onto the Arithmetic Mean. Its improvement degradation relative to the ideal Geometric Mean projection is bounded by the empirical diversity of the sample group:*

$$\text{Improv}(\widehat{\pi}_{Hybrid}) \geq \text{Improv}(\widehat{\pi}_{Ideal}) \\ - 2LB \, \mathbb{E}_x\left[H^2(\{\nu_k(\cdot|x)\})\right],$$

*where $L$ is a Lipschitz constant and $B$ upper bounds the gradient norms of $\log \pi(y)$.*

This result (proven in Appendix F.4) guarantees that shifting from the geometric (ideal) to the arithmetic mean incurs only a small loss when the sample does not admit too large a diversity.

**2. Majority Vote.** While the Arithmetic Mean is stable, averaging distributions inherently increases entropy (Cover & Thomas, 2006). For reasoning tasks where a single correct answer exists, we desire a sharpened target to drive the policy toward decisive self-consistency. Therefore, our practical algorithm adopts the *Majority Vote* (Mode), which can be viewed as the arithmetic mean followed by a hard $\arg\max$ operator:

$$\mathsf{q}^*(z) = \mathbb{1}(z = \arg\max \mathsf{q}_{\mathsf{AM}}(z)).$$

This choice sacrifices the granular probability information of the Arithmetic Mean in exchange for a lower-entropy target that drives the policy toward decisive self-consistency.

In Appendix F, we present a more extensive discussion of the properties of different Target choices.

### 4.3. COHERENCE-GRPO Algorithm

Our generic GRPO-based algorithm, GAME-GRPO, can be instantiated in the case of the *Coherence Game* to provide an algorithm tackling tasks with a discrete answer space $\mathcal{Z}$, such as mathematical reasoning.

**Target Step (Orbit Consensus).** To enforce coherence, the Target $\mathsf{q}^*$ must represent the consensus not just of the model's current output for $x$, but of the entire equivalence class Orbit$(x)$. Computationally, we approximate this by sampling a small set of inputs from the orbit (e.g., $x$ and a paraphrase $x' = \Phi(x)$). We generate groups of traces for both, pool their answers, and compute the Global Mode of this pooled set. The reward for a trace $y$ given $x$ is then determined by whether it matches this *global* consensus $z^*$, rather than merely the local consensus of $x$. This penalizes the model if $\pi(\cdot|x)$ and $\pi(\cdot|x')$ disagree, directly optimizing the coherence objective.

**Reward Definition.** Based on the analysis in Section 4.2, we adopt the **Majority Vote** (Mode) target to enforce sharpening. This sets $\mathsf{q}^*$ to be a Dirac distribution centered on the most frequent answer $z^*$. The reward is derived directly from this sharpened target (setting $R_i = 1$ if $\mathsf{E}(y_i) = \text{Mode}(\{\mathsf{E}(y_j)\})$ and $0$ otherwise).

### 4.4. Pairwise Answer-Level Coherence

The previous discussion supposes the existence of a deterministic extraction function $\mathsf{E}(y) \to z$ that maps a reasoning trace to a canonical discrete answer. However, in open-ended domains (e.g., question answering, creative generation), semantically equivalent answers often differ in surface form, making exact-match consensus brittle. To ad-

---

**Algorithm 1** COHERENCE-GRPO (Orbit-Level Consensus)

**Input:** Dataset $\mathcal{D}$, Policy $\pi$, Extractor $\mathsf{E}$, Transformation $\Phi$ (e.g., paraphrase).

1: **for** each training step **do**
2:     Sample batch of seeds $B \sim \mathcal{D}$.
3:     **for** each seed $x \in B$ **do**
4:         **1. Orbit Sampling:**
5:         Generate orbit inputs $\mathcal{O}_x = \{x, \Phi(x)\}$.    // *e.g., Original + Paraphrase*
6:         Sample $K$ traces for *each* input in $\mathcal{O}_x$:
7:         $G = \{(y_{i,j}, x^{(j)}) \mid x^{(j)} \in \mathcal{O}_x, i \in \{1..K\}, y_{i,j} \sim \pi(\cdot|x^{(j)})\}$.
8:         **2. Target Step (Orbit Consensus):**
9:         Extract all answers: $\mathcal{Z}_G = \{\mathsf{E}(y) \mid (y, \cdot) \in G\}$.
10:       Compute *Global Consensus* across the orbit:
11:       $z^* = \text{Mode}(\mathcal{Z}_G)$.
12:       **3. Reward Calculation:**
13:       For each trace $y_{i,j}$ from input $x^{(j)}$:
14:       $R_{i,j} = \mathbb{1}(\mathsf{E}(y_{i,j}) = z^*)$.   // *Reward match to GLOBAL consensus*
15:       Compute advantages $A_{i,j}$ (standardized within the orbit group $G$).
16:     **end for**
17:     **4. Update:** Optimize $\pi$ using GRPO on all generated traces.
18: **end for**

---

dress this, we develop in Appendix B a relaxed formulation, PAIRWISE-GRPO (Algorithm 2), that replaces the extraction function with a pairwise semantic distance $d(y, y')$. The reward for a trace is its average agreement with the orbit group, enabling robust consensus even when answers are string-distinct but semantically equivalent.

## 5. Experimental results

We evaluate our framework in two stages. First, we validate the variance reduction properties of the GAME-GRPO estimator in a controlled synthetic environment with explicit ground-truth heterogeneity. Second, we scale the approach to Answer-Level Self-Improvement on large language models, demonstrating significant gains on mathematical reasoning (GSM8K) and open-ended question answering (TriviaQA).

### 5.1. Synthetic Validation: Variance Reduction in Many-to-One Mappings

Before scaling to large language models, we validate our theoretical claims in a controlled synthetic environment. A core challenge in ALFT is the "many-to-one" nature of reasoning, where thousands of latent traces $y$ map to a single answer $z$ (Uesato et al., 2022). Standard gradient estimators

| Algorithm | Model | Baseline ACC (95% CI) | Our ACC (95% CI) | Abs. ↑ (pp) | Rel. ↑ |
|---|---|---|---|---|---|
| Pairwise-GRPO | Qwen-3B | 75.06 [72.65, 77.32] | 79.61 [77.35, 81.69] | +4.55 | +6.06% |
| | Llama | 66.19 [63.59, 68.69] | 72.71 [70.24, 75.04] | +6.52 | +9.85% |
| | Phi-3 | 73.69 [71.25, 76.00] | 82.87 [80.74, 84.80] | +9.18 | +12.46% |
| Coherence-GRPO | Qwen-3B | 75.06 [72.65, 77.32] | 80.36 [78.13, 82.42] | +5.30 | +7.06% |
| | Llama | 66.19 [63.59, 68.69] | 69.37 [66.83, 71.80] | +3.18 | +4.80% |
| | Phi-3 | 73.69 [71.25, 76.00] | 81.50 [79.32, 83.50] | +7.81 | +10.60% |

*Table 1.* GSM8K improvement (greedy decoding). Absolute improvement is in percentage points (pp); relative improvement is computed as (Our ACC – base)/base.

| Algorithm | Model | Base EM | Our EM | Rel. ↑ (EM) | Base F1 (95% CI) | Our F1 (95% CI) | Rel. ↑ (F1) |
|---|---|---|---|---|---|---|---|
| Pairwise | Qwen | 32.95 | 35.32 | +7.19% | 39.70 ± 0.67 | 40.28 ± 0.69 | +1.46% |
| | Llama | 39.12 | 47.85 | +22.32% | 48.56 ± 0.66 | 53.52 ± 0.70 | +10.22% |
| | Phi-3 | 32.03 | 45.50 | +42.06% | 42.76 ± 0.63 | 50.51 ± 0.70 | +18.12% |
| Coherence | Qwen | 32.95 | 32.90 | -0.15% | 39.70 ± 0.67 | 39.62 ± 0.67 | -0.20% |
| | Llama | 39.12 | 40.25 | +2.89% | 48.56 ± 0.66 | 49.35 ± 0.66 | +1.63% |
| | Phi-3 | 32.03 | 32.00 | -0.09% | 42.76 ± 0.63 | 42.79 ± 0.63 | +0.07% |

*Table 2.* TriviaQA results. Relative improvement is computed as (ours – base)/base. For F1, we report the 95% CI half-width as ±. For algorithms, Pairwise and Coherence stand for Pairwise-GRPO and Coherence-GRPO, respectively.

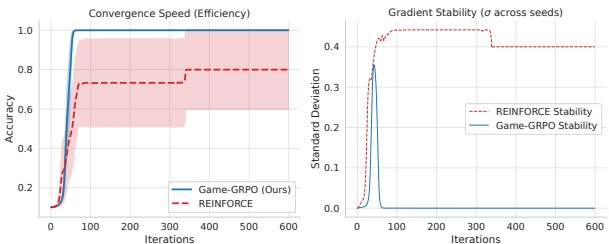

*Figure 2.* Robustness to Extreme Heterogeneity. We compare GAME-GRPO (Blue) against REINFORCE with a global baseline (Red) on a 1000-to-1 redundant mapping task. The environment switches between "Easy" (Bias 8.0) and "Hard" (Bias 0.0) instances. *(Left)* GAME-GRPO isolates the true advantage signal, converging to 1.0 accuracy. The baseline is overwhelmed by the 8× bias, fluctuating around 0.8. *(Right)* The stability plot reveals the mechanism: the global baseline induces massive gradient variance ($\sigma \approx 0.4$), whereas the group-relative baseline reduces variance to near zero, enabling deterministic-like convergence.

struggle to assign credit in this setting, especially when the task difficulty varies per instance (heterogeneous difficulty).

**Experimental Setup.** We construct a "Redundant Mapping" environment with a trace space $|\mathcal{Y}| = 10,000$ mapping to an answer space $|\mathcal{Z}| = 10$ (a $1000 : 1$ redundancy ratio). To simulate the extreme variance of real-world prompts, we introduce a stochastic difficulty bias. At each step, the environment is instantiated as either "Easy" (Base Reward = 8.0) or "Hard" (Base Reward = 0.0). A correct answer yields a marginal signal of +1.0. This creates a signal-to-Bias ratio of 1:8, meaning the environmental noise is eight times larger than the learning signal.

**Baselines.** We compare GAME-GRPO against a strong RE-INFORCE baseline equipped with a global moving-average baseline.

**Results.** Figure 2 illustrates the learning dynamics.

GAME-GRPO (Blue) effectively decouples the signal from the difficulty bias by computing the baseline dynamically from the group, converging rapidly to optimal performance (100% accuracy) with negligible variance. In contrast, the REINFORCE baseline (Red) fails to normalize the extreme heterogeneity: because the global average ($\approx 4.0$) is too low for "Easy" tasks and too high for "Hard" tasks, the estimator suffers from catastrophic variance ($\sigma \approx 0.4$) and plateaus at sub-optimal performance ($\approx 80\%$), unable to fully resolve the credit assignment problem. This confirms that the GAME-GRPO baseline derived in Section 3 is not merely an efficiency trick, but a prerequisite for stability in heterogeneous reasoning tasks.

### 5.2. Answer-Level Coherence on LLMs (GSM8K & TriviaQA)

We evaluate answer-level self-improvement via coherence on GSM8K and TriviaQA datasets. We use the official train split to construct paraphrase groups for training and report performance on the official test split with 1319 questions. Each example is formatted as an instruction-following chat prompt with a fixed system message and a user message containing the math word problem.

We test `Qwen2.5-3B-Instruct`, `Phi-3-mini-4k-instruct`, and `Llama-3.2-3BInstruct`, referred to as `Qwen`, `Phi3`, and `Llama`, respectively. We generate one paraphrase for each question in the training set with `Qwen` and greedy decoding.

For all models, we fine-tune with QLoRA (4-bit NF4 quantization with double-quantization; compute dtype bfloat16 on GPU) and attach a LoRA adapter with $r = 8$, $\alpha = 16$, dropout 0.05 (target modules auto-detected among the usual projection layers).

For GSM8K, training uses Pairwise-GRPO on paired prompts (original + paraphrase) with 2 pairs per step and 32 rollouts per prompt. We set KL weight to 0.02, learning rate to $1e-5$, and max completion length to 512. We train for 3200 steps, disable batch dispatch/splitting in the accelerator. We report the results after 3200 steps in Table 1. Due to computation constraints, we use 1 pair per step and 24 rollouts per prompt for `Phi3`.

For TriviaQA, we trained with the same parameters. We additionally instantiate an embedder `SentenceTransformer` on GPU when available, and use its embeddings to judge semantic similarity between generated answers. We test and report the improvement with GRPO in Table 2.

**Analysis: GSM8K.** On GSM8K (Table 1), both PAIRWISE-GRPO and COHERENCE-GRPO yield consistent and substantial improvements across all three models, with absolute gains ranging from +3.18 to +9.18 percentage points. The uniformity of these gains confirms that the game-theoretic reward signal—derived from the orbit-level consensus—provides a principled and stable training signal for discrete reasoning tasks where a deterministic extraction function E is available.

**Analysis: TriviaQA.** On TriviaQA (Table 2), the two algorithms exhibit sharply divergent behavior. PAIRWISE-GRPO delivers large improvements (up to +42.06% relative EM for Phi-3), whereas COHERENCE-GRPO produces negligible or even slightly negative changes. This gap is explained by the nature of the task: TriviaQA is an open-ended QA benchmark where semantically equivalent answers frequently differ in surface form (e.g., "NYC" vs. "New York City"). COHERENCE-GRPO relies on a deterministic extractor E to compute the answer mode, which cannot recognize such equivalences and consequently produces noisy consensus targets. In contrast, PAIRWISE-GRPO bypasses this bottleneck entirely by using a learned semantic distance metric $d(y, y')$, enabling it to cluster and reinforce semantically equivalent answers even when their surface strings differ.

**Practical guidance.** These results suggest that COHERENCE-GRPO is preferred when a reliable extraction function exists (e.g., numerical answers, multiple-choice), while PAIRWISE-GRPO should be used for open-ended domains where answer equivalence is semantic rather than syntactic. Theoretical guarantees (Theorem 2.1) hold for both; the practical difference lies in the consensus signal quality.

**Convergence and sample coverage.** Theorem F.2 (Appendix F.4) shows that using the empirical frequency estimator (Arithmetic Mean) in place of the ideal geometric mean degrades theoretical improvement by at most a factor

| Metric | Base | S-GRPO | $\Delta$ |
|---|---|---|---|
| Benign Non-Refusal | 1.000 | 0.990 | $-0.010$ |
| Benign Refusal | 0.000 | 0.010 | $+0.010$ |
| Harmful Toxicity | 0.0733 | 0.0677 | $-0.0056$ |
| Harmful Refusal | 0.065 | 0.070 | $+0.005$ |
| Harmful Toxic @0.5 | 0.055 | 0.050 | $-0.005$ |

*Table 3.* SAFETY-GRPO pilot on `Qwen2.5-3B`. The primal-dual algorithm enforces safety budgets: it reduces harmful toxicity while preserving benign helpfulness.

proportional to the Generalized Squared Hellinger distance of the sample group, guaranteeing monotonic improvement for moderate group sizes ($K = 24$ or $32$).

### 5.3. Broader Empirical Validation

**Diversity.** For the diversity objective, favorable experimental results have already been reported by Li et al. (2025), who empirically demonstrated that the Inverse-Frequency Reward effectively combats mode collapse in reasoning tasks. Our framework provides the theoretical grounding for these results: as shown in Appendix C, the Inverse-Frequency heuristic is exactly the GRPO update step for the Maximum Entropy Distributional Alignment Game (Algorithm 3).

**Safety.** To validate the Safety instantiation (Appendix D), we conducted experiments with SAFETY-GRPO (Algorithm 4) and report the results in Section H. We fine-tuned `Qwen2.5-3B` with LoRA to optimize for helpfulness subject to dual safety constraints: harmful toxicity budget < 0.10 and benign refusal budget < 0.10. As a demonstration, Table 3 confirms that the primal-dual update behaves as theoretically predicted: SAFETY-GRPO successfully reduces harmful toxicity and the toxic generation rate while maintaining a $99\%$ benign non-refusal rate.

## 6. Conclusion

We introduced *Distributional Alignment Games*, a variational framework that lifts Answer-Level Fine-Tuning from an intractable marginalization problem to a tractable game between a Policy and a Target distribution. This perspective provides a unified theoretical lens: distinct objectives such as diversity, coherence, and safety, are merely different strategies played by the Target, grounding recent heuristic successes in convex duality theory. We bridged theory and practice by deriving scalable algorithms (COHERENCE-GRPO, PAIRWISE-GRPO) whose rewards follow directly from the game equilibrium, and extended the framework to open-ended domains and distributional safety constraints. A discussion of limitations is provided in Appendix A.

## Impact Statement

This paper aims to advance machine learning by providing a game-theoretic framework for answer-level fine-tuning and distributional alignment. The methods studied here may improve the reliability, coherence, and safety of language models by optimizing distributional properties of their outputs rather than only individual generations.

Our empirical safety pilot uses publicly available safety datasets, including harmful prompts from BeaverTails and benign prompts from XSTest, to evaluate whether SAFETY-GRPO can reduce toxic generations while preserving benign non-refusal. These experiments are intended for research evaluation of safety training methods, not for deployment of a production safety system. Because the method fine-tunes models on harmful-prompt data, care should be taken to avoid releasing models or artifacts that increase harmful capabilities. More broadly, distributional alignment methods could be misused if the target distribution is chosen to optimize undesirable behaviors; appropriate dataset curation, evaluation, and deployment safeguards remain important.

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

# Contents of Appendix

## A. Limitations

Our experiments focus on 3B-class models and two benchmarks; validation at larger scales is needed. The practical algorithms approximate the ideal consensus target (e.g., majority vote vs. geometric mean); although Theorem F.2 bounds this gap, highly multimodal settings deserve further study. Finally, convergence guarantees (Appendix E) assume exact subproblem solutions, whereas implementations use stochastic GRPO updates.

## B. Pairwise Answer-Level Coherence

In previous sections, we assumed the existence of a deterministic extraction function $\mathsf{E}(y) \to z$ that maps a reasoning trace to a canonical discrete answer. In many complex domains, such as open-ended question answering, summarization, or semantic code retrieval, such a function is difficult to define or brittle to implement. A parser may fail to extract an answer even when the reasoning is correct, or two string-distinct answers might be semantically equivalent. Even for some math problems, it may difficult to come up with an accurate extraction function.

However, in these domains, it is often feasible to define a *pairwise disagreement function* $\mathsf{d}(y, y') \in [0, 1]$ that quantifies the semantic distance between two traces. $\mathsf{d}(y, y') = 0$ implies perfect consistency, while $\mathsf{d}(y, y') = 1$ implies total contradiction.

Note that the answer-level formulation via an extraction function $\mathsf{E}$ can always be cast as a pairwise one using $\mathsf{d}(y, y') = \mathbb{I}(\mathsf{E}(y) \neq \mathsf{E}(y'))$.

We regard the pairwise formulation as a critical extension of the standard framework, as it adapts the Coherence Game to open-ended domains where exact answer matching is infeasible. Moreover, this pairwise perspective naturally generalizes to other answer-level games.

### B.1. Problem Formulation

The goal of self-improvement is to ensure that the model's outputs are semantically consistent across task-preserving transformations. We therefore define the pairwise incoherence penalty over the orbit of an input $x$:

$$\text{Incoherence}(\pi_\theta) = \mathbb{E}_{x' \sim \text{orbit}(x)} \left[ \mathbb{E}_{y \sim \pi(\cdot|x), y' \sim \pi(\cdot|x')} [d(y, y')] \right]. \tag{9}$$

This forces the distribution $\pi(\cdot|x)$ to be semantically aligned with $\pi(\cdot|x')$, rather than just collapsing to a low-entropy state for a single prompt.

### B.2. PAIRWISE-GRPO Algorithm

Target Step (Best Response). Since we cannot compute a mode, we view the Target $\mathsf{q}^*$ as the "center of mass" of the semantic distribution. We define a pairwise distance metric $d(y, y')$. The optimal Target assigns high probability to traces that minimize the average transport cost to all other traces in the group.

Reward Definition. The reward is the *centrality* of the trace, which proxies for $\log \mathsf{q}^*$:

$$R_i = \frac{1}{K} \sum_{j=1}^{K} (1 - d(y_i, y_j)).$$

Thus, high $R_i$ implies $y_i$ is close to the group consensus $\mathsf{q}^*$.

**Proposition B.1** (Derivation of Orbit-Centrality). *Let the orbit-level incoherence objective be defined as the expected pairwise distance between traces generated from inputs in the same equivalence class $\mathcal{O}_x$:*

$$\mathcal{J}_{orbit}(\pi) = \mathbb{E}_{x^{(a)}, x^{(b)} \sim \mathcal{O}_x} \left[ \mathbb{E}_{y_a \sim \pi(\cdot|x^{(a)}), y_b \sim \pi(\cdot|x^{(b)})} [d(y_a, y_b)] \right].$$

*The gradient of this objective with respect to the policy parameters $\theta$ satisfies:*

$$\nabla_\theta \mathcal{J}_{orbit}(\pi) = 2 \mathbb{E}_{x \sim \mathcal{O}_x} \left[ \mathbb{E}_{y \sim \pi(\cdot|x)} \left[ \nabla_\theta \log \pi(y|x) \underbrace{\mathbb{E}_{x' \sim \mathcal{O}_x} \left[ \mathbb{E}_{y' \sim \pi(\cdot|x')} [d(y, y')] \right]}_{\textit{Orbit Incoherence}} \right] \right]. \tag{10}$$

*To minimize this objective via policy gradient, the optimal reward signal $R^*(y)$ must be proportional to the negative of the gradient coefficient. Thus, we define the reward as the negative expected distance to the orbit:*

$$R^*(y) \propto - \mathop{\mathbb{E}}_{x' \sim \mathcal{O}_x} \left[ \mathop{\mathbb{E}}_{y' \sim \pi(\cdot | x')} [d(y, y')] \right].$$

*Proof.* Let the objective for a fixed pair of inputs $(x^{(a)}, x^{(b)})$ be $J_{a,b}(\pi) = \mathbb{E}_{y_a \sim \pi_a, y_b \sim \pi_b}[d(y_a, y_b)]$, where $\pi_k = \pi(\cdot | x^{(k)})$. We compute the gradient using the log-derivative trick:

$$\begin{aligned}
\nabla_\theta J_{a,b}(\pi) &= \nabla_\theta \sum_{y_a, y_b} \pi_a(y_a) \pi_b(y_b) d(y_a, y_b) \\
&= \sum_{y_a, y_b} (\pi_a(y_a)[\nabla \log \pi_a(y_a)] \pi_b(y_b) + \pi_a(y_a) \pi_b(y_b)[\nabla \log \pi_b(y_b)]) d(y_a, y_b) \\
&= \mathop{\mathbb{E}}_{y_a, y_b} [\nabla \log \pi_a(y_a) \cdot d(y_a, y_b)] + \mathop{\mathbb{E}}_{y_a, y_b} [\nabla \log \pi_b(y_b) \cdot d(y_a, y_b)].
\end{aligned}$$

The total gradient is the sum over all pairs $a, b$. By symmetry, the total contribution for a specific trace $y$ generated from input $x$ appears twice in the summation (once as $y_a$, once as $y_b$). Thus, the gradient coefficient for $\nabla \log \pi(y|x)$ is exactly:

$$2 \times \sum_{x' \in \mathcal{O}_x} \mathop{\mathbb{E}}_{y' \sim \pi(\cdot | x')} [d(y, y')].$$

We define the reward $R(y)$ to be proportional to the negative of this term. The factor of 2 is a constant scaler that is absorbed by the learning rate or advantage standardization in GRPO. □

---

**Algorithm 2** PAIRWISE-GRPO (Orbit-Level Consistency)

---

**Input:** Dataset $\mathcal{D}$, Group size $K$, Policy $\pi$, Metric $d$, Transformation $\Phi$.

1: **for** each training step **do**
2:     Sample batch of seeds $B \sim \mathcal{D}$.
3:     **for** each seed $x \in B$ **do**
4:         **1. Orbit Sampling:**
5:         Generate orbit inputs $\mathcal{O}_x = \{x, \Phi(x)\}$.
6:         Sample $K$ traces for each input:
7:         $G = \{(y_{i,j}, x^{(j)}) \mid x^{(j)} \in \mathcal{O}_x, i \in \{1..K\}, y_{i,j} \sim \pi(\cdot | x^{(j)})\}$.
8:         **2. Target Step (Orbit Centrality):**
9:         *// Reward is centrality relative to the ENTIRE orbit group*
10:        **for** each trace $y \in G$ **do**
11:          $R(y) = \frac{1}{|G|} \sum_{y' \in G} (1 - d(y, y'))$.
12:        **end for**
13:        **3. Advantage Calculation:**
14:        Compute standardized advantages $A_k$ across the orbit group $G$.
15:     **end for**
16:     **4. Update:** Optimize $\pi$ using GRPO with advantages $A_k$.
17: **end for**

---

The code is available at
https://colab.research.google.com/drive/1sLzNyZEAFVYm5GkLRSy-CJ8IM5l8rjsd?usp=sharing.

## C. Diversity-Promotion: Problem and Algorithms

In domains like creative generation or exploratory search, the goal is to prevent mode collapse and encourage the policy to cover a broad distribution of valid answers, that is *diversity*.

## C.1. The Inverse-Frequency Heuristic

A prominent recent approach to this problem is the *Inverse-Frequency Reward* proposed by Li et al. (2025). To combat mode collapse, they introduce a heuristic reward that penalizes frequent answers within a sampled group.

Formally, given a group of traces $G = \{y_1, \ldots, y_K\}$ generating answers $z_i = \mathsf{E}(y_i)$, they define the reward for a trace $y_i$ as the negative log-frequency of its answer:

$$R_{\text{inv-freq}}(y_i) = -\log\left(\frac{\text{Count}(z_i)}{K}\right). \tag{11}$$

Intuitively, this mechanism assigns high rewards to rare answers (high surprisal) and low rewards to common ones, pushing the policy to spread its mass to the tails of the distribution. While empirically effective, this method was originally motivated primarily by intuition regarding exploration.

## C.2. Theoretical Grounding via Distributional Alignment

Our framework provides a rigorous theoretical derivation for this algorithm. Within the ALFT framework, diversity is not a heuristic but a specific instantiation of the objective function. We will refer to the resulting algorithm by DIVERSITY-GRPO.

### C.2.1. RECOVERING HEURISTIC GRPO-BASED ALGORITHM

Let the answer-level regularization functional be the entropy functional, $\mathcal{R}(\nu) = \lambda H(\nu)$. The gradient of the entropy with respect to the marginal distribution is:

$$\nabla H(\nu_\pi) = -\mathop{\mathbb{E}}_{z \sim \nu_\pi}\left[\nabla \log \nu_\pi(z)(\log \nu_\pi(z) + 1)\right].$$

We now derive the gradient of the entropy functional $\mathcal{R}(\nu_\pi) = H(\nu_\pi)$ with respect to the policy parameters. Let $\nu_\pi(z)$ denote the marginal probability of answer $z$, then we can write:

$$
\begin{aligned}
\nabla H(\nu_\pi) &= \nabla\left(-\sum_{z \in \mathcal{Z}} \nu_\pi(z) \log \nu_\pi(z)\right) \\
&= -\sum_{z \in \mathcal{Z}} \nabla(\nu_\pi(z) \log \nu_\pi(z)) \\
&= -\sum_{z \in \mathcal{Z}} \nabla\nu_\pi(z)(\log \nu_\pi(z) + 1) \\
&= -\sum_{z \in \mathcal{Z}} \nu_\pi(z)\nabla \log \nu_\pi(z)(\log \nu_\pi(z) + 1) \qquad \text{(log-derivative identity: } \nabla\nu = \nu\nabla\log\nu\text{)} \\
&= -\mathop{\mathbb{E}}_{z \sim \nu_\pi}\left[\nabla \log \nu_\pi(z)(\log \nu_\pi(z) + 1)\right].
\end{aligned}
$$

This derivation shows the implicit reward structure of the maximum entropy objective. Comparing this expression to the standard policy gradient form $\nabla J = \mathbb{E}[\nabla \log \pi \cdot R]$, we identify the effective reward signal $R(z)$ as the term scaling the gradient:

$$R(z) = -(\log \nu_\pi(z) + 1) \approx -\log \nu_\pi(z).$$

Since $\log \nu_\pi(z)$ is negative, the term $-\log \nu_\pi(z)$ is positive. This means that answers with low probability (rare answers) generate a large positive reward, while answers with high probability (common answers) generate a smaller reward. Optimizing this objective therefore incentivizes the policy to shift probability mass from frequent answers to rare ones, theoretically justifying the *Inverse-Frequency* heuristic used for diversity promotion.

This implies that to maximize entropy via stochastic gradient descent, the reward signal for a trace $y$ leading to answer $z$ must be proportional to its *surprisal* (negative log-likelihood):

$$R_{\text{theory}}(y) \propto -\log \nu_\pi(\mathsf{E}(y)).$$

To implement this in the GRPO setting, we must estimate the current marginal $\nu_\pi$ from the sampled group $G$. Using the Arithmetic Mean (empirical frequency) estimator discussed in Section F, we have $\widehat{\nu}_\pi(z) \approx \frac{\text{Count}(z)}{K}$. Substituting this into the theoretical reward yields:

$$R_{\text{theory}}(y_i) \approx -\log\left(\frac{\text{Count}(\mathsf{E}(y_i))}{K}\right).$$

This demonstrates that the Inverse-Frequency Reward of Li et al. (2025) is exactly a valid GRPO update step for the *Max-Entropy Distributional Alignment Game*.

### C.2.2. ALTERNATIVE DIVERSITY-GRPO ALGORITHMS

The game-theoretic perspective reveals that the inverse-frequency reward is just one of many possible diversity-promoting objectives. By changing the functional $\mathcal{R}$, we can derive alternative algorithms instances of our general DIVERSITY-GRPO algorithm, that may offer better stability or properties:

1. *Gini Diversity*: If we instead maximize the Gini index $\mathcal{R}(\nu) = 1 - \sum_z \nu(z)^2$ (collision probability), the gradient yields a linear minority reward:

$$R_{\mathrm{gini}}(y_i) = 1 - \frac{\mathrm{Count}(\mathsf{E}(y_i))}{K}.$$

   Unlike the log-reward, this is bounded and does not explode for unique answers (where count is 1), potentially offering more stable optimization for small group sizes.

2. *Targeted Exploration*: We can replace the uniform entropy objective with a divergence relative to a specific prior $Q_{\mathrm{prior}}$. The game then drives the policy to match $Q_{\mathrm{prior}}$ rather than just spreading mass uniformly. The reward becomes $R(y_i) = \log Q_{\mathrm{prior}}(\mathsf{E}(y_i)) - \log \widehat{\nu}_\pi(\mathsf{E}(y_i))$, acting as an importance-weighted exploration signal.

---

**Algorithm 3** DIVERSITY-GRPO (General Formulation)

---

**Input:** Dataset $\mathcal{D}$, Policy $\pi$, Group size $K$, **Objective** (Entropy or Gini).

1: **for** each training step **do**
2:     Sample batch $B \sim \mathcal{D}$.
3:     **for** each $x \in B$ **do**
4:         **1. Group Sampling:** Sample $K$ traces $\{y_1, \ldots, y_K\} \sim \pi(\cdot|x)$.
5:         **2. Target Step (Marginal Estimation):**
6:         Estimate empirical marginals: $\widehat{\nu}(z) = \frac{\mathrm{Count}(z)}{K}$.
7:         **3. Reward Calculation:**
8:         **if Objective** is Entropy (Li et al., 2025) **then**
9:             $R_k = -\log \widehat{\nu}(\mathsf{E}(y_k))$     *// Inverse-Frequency (High Variance)*
10:       **else if Objective** is Gini **then**
11:           $R_k = 1 - \widehat{\nu}(\mathsf{E}(y_k))$     *// Probability of Collision (Bounded)*
12:       **end if**
13:       Compute advantages $A_k$ via Eq. (5).
14:     **end for**
15:     **4. Update:** Optimize $\pi$ using GRPO.
16: **end for**

---

Algorithm 3 illustrates the generality of our framework. By selecting the Entropy objective, we recover the inverse-frequency method of Li et al. (2025). However, our game-theoretic derivation reveals that we can easily substitute the Gini objective to obtain a bounded, lower-variance reward signal $(1 - \widehat{\nu})$ without changing the underlying algorithm.

## D. Distributional Constraint Satisfaction: Safety and Fairness

While the Diversity and Coherence games optimize for specific shapes of the distribution (spread vs. point-mass), a third critical class of problems involves *Distributional Constraints*. In real-world deployment, we often require the aggregate behavior of the model to satisfy safety or fairness guarantees, even if individual answers are technically correct.

Here are some examples for both types: (1) Safety/Toxicity: We may require that the expected toxicity score of generated answers remains below some threshold $\epsilon$, or that the rate of refusal for benign prompts is minimized; (2) Fairness: A medical diagnosis model may be required to yield positive predictions at equal rates across demographic groups (Demographic Parity). This is a constraint on the marginal distribution of answers $\nu_\pi$, not on any single trace.

## D.1. Problem Formulation

We formulate this as a constrained optimization problem. We wish to minimize the divergence from the reference policy $\pi_0$ subject to the marginal distribution $\nu_\pi$ remaining within a valid set $\mathcal{C}$:

$$\min_{\pi \in \Pi} \quad D_{\mathrm{KL}}(\pi \| \pi_0) \quad \text{s.t.} \quad \nu_\pi \in \mathcal{C}. \tag{12}$$

Typically, $\mathcal{C}$ is defined by linear constraints on feature functions $\mathbf{c}(z)$ (e.g., toxicity classifiers or demographic indicators):

$$\mathcal{C} = \left\{ \nu \in \Delta(\mathcal{Z}) \mid \mathop{\mathbb{E}}_{z \sim \nu}[\mathbf{c}(z)] \le \mathbf{b} \right\}.$$

In our ALFT framework, this maps to the general problem (1) by defining the answer-level functional $\mathcal{R}(\nu)$ as the convex indicator function of the set $\mathcal{C}$ (taking value 0 if $\nu \in \mathcal{C}$ and $+\infty$ otherwise).

## D.2. Game Interpretation: The Projection Game

In the game-theoretic dual, the role of the Target q is to act as the *Corrected Distribution*. Since $\mathcal{R}$ is the indicator of $\mathcal{C}$, the optimal Target $\mathsf{q}^*$ in the game becomes the Information Projection (I-Projection) of the current policy's marginal $\nu_\pi$ onto the constraint set $\mathcal{C}$.

$$\mathsf{q}^* = \operatorname*{argmin}_{\mathsf{q} \in \mathcal{C}} D_{\mathrm{KL}}(\mathsf{q} \| \nu_\pi). \tag{13}$$

Intuitively, the Target looks at the current noisy or unsafe behavior of the policy ($\nu_\pi$) and finds the closest distribution that satisfies the safety/fairness constraints. The Policy then updates to mimic this safe Target.

## D.3. SAFETY-GRPO algorithm

Computing the exact projection $\mathsf{q}^*$ is difficult. However, for linear constraints $\mathbb{E}[\mathbf{c}] \le \mathbf{b}$, the projection has a known exponential form parametrized by Lagrange multipliers $\lambda \in \mathbb{R}_{\ge 0}^d$:

$$\mathsf{q}^*(z) \propto \nu_\pi(z) \exp\!\left(-\lambda^\top \mathbf{c}(z)\right).$$

Substituting this target into the GRPO reward definition ($R(y) = \log \mathsf{q}^*(\mathsf{E}(y))$), we derive a computationally efficient algorithm based on Dual Ascent.

**Reward Structure.** The effective reward for a trace $y$ is simply the negative weighted constraint violation:

$$R_{\text{safe}}(y) = -\lambda^\top \mathbf{c}(\mathsf{E}(y)). \tag{14}$$

Here, $\lambda$ acts as a dynamic penalty weight. The algorithm alternates between two steps:

1. Policy Step (GRPO): Update $\pi$ using rewards $R_{\text{safe}}(y)$ to minimize constraint violations.

2. Dual Step (Update $\lambda$): Update the penalty weights $\lambda$ via gradient ascent to enforce the constraints strictly:

$$\lambda \leftarrow \max\!\left(0, \lambda + \alpha\!\left(\mathop{\mathbb{E}}_{z \sim \pi}[\mathbf{c}(z)] - \mathbf{b}\right)\right).$$

This derivation further shows that standard approaches to constrained RL (like Lagrangian relaxation) are, in fact, finding the Nash Equilibrium of the Distributional Alignment Game where the Target plays the strategy of the *closest safe distribution*.

This leads to a theoretically grounded algorithm we refer to as SAFETY-GRPO.

---

**Algorithm 4** SAFETY-GRPO (Primal-Dual)

---

**Input:** Dataset $\mathcal{D}$, Policy $\pi$, Constraints $\mathbf{c}(z) \leq \mathbf{b}$, Step size $\alpha$.

1: Initialize Lagrange multipliers $\lambda \leftarrow \mathbf{0}$.
2: **for** each training step **do**
3:     Sample batch $B \sim \mathcal{D}$.
4:     **for** each $x \in B$ **do**
5:         Sample $K$ traces $\{y_1, \ldots, y_K\} \sim \pi(\cdot|x)$.
6:         Extract answers $z_k = \mathsf{E}(y_k)$ and features $\mathbf{c}(z_k)$.
7:         **Target Step:** Implicitly defined by penalty weights $\lambda$.
8:         **Reward Calculation:**
9:           $R_k = -\lambda^\top \mathbf{c}(z_k)$     *// Penalize constraint violations*
10:        Compute advantages $A_k$ via Eq. (5).
11:     **end for**
12:     **Policy Step:** Update $\pi$ using GRPO with advantages $A_k$.
13:     **Dual Step:** Update penalties via gradient ascent:
14:        $\widehat{\mathbf{c}} = \frac{1}{|B|K} \sum_{x \in B} \sum_{k=1}^{K} \mathbf{c}(z_k)$    *// Estimate expected violation*
15:        $\lambda \leftarrow \max(0, \lambda + \alpha(\widehat{\mathbf{c}} - \mathbf{b}))$.
16: **end for**

---

## E. Solving the Minimax Game via No-Regret or Best Response Dynamics

Crucially, the game formulation is computationally tractable. It offers a flexible theoretical basis that admits two distinct types of solutions, depending on the computational constraints and the properties of the functional $\mathcal{R}$.

**1. General Solution via No-Regret Dynamics.** For some objectives, the game permits a convex-concave parametrization, allowing us to use techniques from no-regret learning to efficiently compute a Nash equilibriumm. Specifically, the objective $\mathcal{G}(\pi, \mathsf{q})$ satisfies the following properties:

- **Strictly Convex in $\pi$**: The term $\mathsf{D}_{\mathrm{KL}}(\pi \| \pi_0)$ is strictly convex in $\pi$, while the coupling term $\mathbb{E}_{y \sim \pi}[\log \mathsf{q}(\mathsf{E}(y))]$ is linear in $\pi$.

- **Concave in Dual Variables**: The objective is concave with respect to the natural dual parameters $u$ (where $u = -\beta \log \mathsf{q}$). While the composition with the non-linear mapping $u \to \mathsf{q}$ does not guarantee concavity in $\mathsf{q}$ for general functionals, the underlying convexity of $\mathcal{R}^*$ ensures the problem is well-posed in the dual space. For specific regularizers (e.g., entropy), concavity in $\mathsf{q}$ may also hold on the simplex.

This structure technically allows for solution via standard no-regret dynamics (e.g., Mirror Descent or Follow-the-Regularized-Leader) (Freund & Schapire, 1999; Shalev-Shwartz, 2012) on the dual variables $u$. By updating $\pi$ to minimize regret and $u$ to maximize regret, the average iterates converge to the Nash Equilibrium at a rate of $O(1/\sqrt{T})$. This provides a robust fallback for cases where exact maximization is difficult.

**2. Efficient Solution via Alternating Best Response.** In the context of ALFT, we can adopt a faster strategy: *Alternating Best Response*. Instead of incremental gradient steps, we solve the inner optimization problems exactly. Crucially, we perform the maximization directly over the target distribution $\mathsf{q}$ rather than the dual variable $u$. This is justified by the bijection established in Section 2.2: finding the optimal $\mathsf{q}^* \in \Delta(\mathcal{Z})$ is equivalent to finding the optimal dual variable $u^*$ (modulo shift) that maximizes the concave dual objective.

We thus iterate the following two steps:

1. **Target Step (Exact Dual Maximization):** Fix $\pi$ and find the optimal target $\mathsf{q}^*$.

$$\mathsf{q}^* = \operatorname*{argmax}_{\mathsf{q} \in \Delta(\mathcal{Z})} \mathcal{G}(\pi, \mathsf{q}).$$

Because of the bijection between $\mathsf{q}$ and the equivalence classes of $u$, this step is equivalent to finding the exact maximizer of the dual objective $u^*$. For many functionals (e.g., entropy or Euclidean distance), this admits a closed-form solution (e.g., centroid).

2. **Policy Step (Exact Primal Minimization):** Fix $q^*$ and update $\pi$ to match the trace-level target $\widetilde{q}(y) \propto \pi_0(y) q^*(E(y))$.

$$\pi_{new} = \underset{\pi \in \Pi}{\arg\min}\, \mathcal{G}(\pi, q^*).$$

This is a standard supervised learning task (KL projection).

**Convergence.** While alternating best response does not converge for general games, it is guaranteed to converge for *strictly convex-concave* games where the subproblems are solved exactly. Since our primal objective $\mathcal{J}(\pi)$ is strictly convex (and the implied dual is strictly concave modulo shifts), this alternating procedure converges to the unique global optimum $(\pi^*, q^*)$ (Tseng, 2001). This justifies our algorithmic choice to iteratively estimate the target and project the policy.

**Connection to Variational EM.** Structurally, our alternating best-response scheme resembles a *Variational Expectation-Maximization (EM)* procedure. The *Target Step* acts as an *E-Step*, inferring the optimal variational distribution $q^*$, while the *Policy Step* acts as an *M-Step*, updating $\pi$ to match these targets. However, unlike standard EM which typically guarantees convergence only to a stationary point, our formulation as a game derived from a convex primal problem allows for stronger global convergence guarantees.

## F. Choice of the Consensus Target

In the Coherence Game, the Target $q^*$ serves as the ground truth estimate that the policy attempts to match. Mathematically, this target is the *Bregman Centroid* of the distributions $\{\nu_k\}_{k=1}^K$ implied by the sampled group. The specific choice of divergence determines the nature of this consensus. We analyze three candidates for $q^*$: the Geometric Mean, the Arithmetic Mean, and the Majority Vote.

### F.1. Geometric Mean (Ideal Consensus)

A. Mathematical Definition. For self-improvement, we seek a sharp consensus that eliminates noise. This is formally captured by minimizing the average KL Divergence:

$$q^* = \underset{q \in \Delta(\mathcal{Z})}{\arg\min}\, \frac{1}{K} \sum_{k=1}^K D_{\mathrm{KL}}(q \| \nu_k).$$

The closed-form solution is the Geometric Mean:

$$q^*(z) = \frac{1}{Z}\left(\prod_{k=1}^K \nu_k(z)\right)^{\frac{1}{K}}, \quad \text{where } Z = \sum_{z' \in \mathcal{Z}}\left(\prod_{k=1}^K \nu_k(z')\right)^{\frac{1}{K}}. \tag{15}$$

B. Approximation Status. This is the exact solution for the strict Coherence objective. It theoretically provides the strongest signal for self-improvement because it enforces *intersectional consistency*: if any trace assigns zero probability to an answer, the consensus probability becomes zero.

C. Computational Properties. Despite its theoretical appeal, the Geometric Mean is *computationally intractable* for two reasons: (1) Normalization: Computing the partition function $Z$ requires summing over the exponentially large space of all possible answers $\mathcal{Z}$; (2) Sample Sparsity: In a GRPO setting, we only observe discrete samples $z_k$, not the full probability vectors $\nu_k$. If we treat samples as one-hot distributions, the product term becomes zero everywhere unless the group is unanimous, rendering the mean degenerate.

D. Statistical Properties. The Geometric Mean has the property of *Veto Power*. It acts as a strict filter, keeping only answers supported by *all* reasoning paths. While this removes hallucinations effectively, it can be overly aggressive in early training, leading to signal collapse if the model is not yet consistent.

### F.2. The Arithmetic Mean (Robust Relaxation)

A. Mathematical Definition. The Arithmetic Mean corresponds to the standard mixture distribution:

$$\overline{\nu}(z) = \frac{1}{K} \sum_{k=1}^K \nu_k(z).$$

It is the unique minimizer for both the *Forward KL Divergence* ($\sum D_{\mathrm{KL}}(\nu_k \| q)$) and the Squared Euclidean Distance.

B. Approximation Status. This is a relaxed solution. While it minimizes a different divergence than the strict mode-seeking Reverse KL, it preserves the support of the distribution.

- **Theoretical Guarantee:** As shown in Appendix F.4, iterative projection onto the Arithmetic Mean guarantees convergence to a coherent equilibrium, provided the initial policy possesses sufficient diversity to cover the true answer.

C. Computational Properties. The Arithmetic Mean is highly tractable. For discrete samples, it is simply the empirical frequency distribution:

$$q^*(z) = \frac{\mathrm{Count}(z \in G)}{K}.$$

This allows for a direct implementation in GRPO using a "Soft Consistency" reward:

$$R_i = \log\left(\frac{\mathrm{Count}(z_i)}{K}\right).$$

This reward is granular, distinguishing between strong consensus (e.g., 90%) and weak consensus (e.g., 40%).

D. Statistical Properties. The Arithmetic Mean enforces *Union Consistency*: it preserves any answer generated by at least one group member. However, it admits a blurring effect issue: a key statistical downside is that averaging distributions *increases* entropy (by Jensen's inequality). While we want the model to sharpen (become more confident), the Arithmetic Mean target effectively asks the model to cover the uncertainty of the group. This blurring must be counteracted by the inherent sharpening of the model's generation process (e.g., $\mathrm{argmax}$ decoding).

### F.3. Majority Vote (Hard Approximation)

A. Mathematical Definition. The Majority Vote (or Mode) is the Dirac distribution centered at the most frequent answer in the group:

$$q^*(z) = \mathbb{I}\left[z = \operatorname*{argmax}_{z'}\left(\frac{1}{K}\sum_{k=1}^{K}\mathbb{I}[z_k = z']\right)\right].$$

B. Approximation Status. This is a *heuristic approximation*. It can be viewed as computing the Arithmetic Mean (step 1) and then applying a hard $\mathrm{argmax}$ operator (step 2) to force the distribution to collapse. This makes it an even coarser approximation than the Arithmetic Mean, discarding all information about secondary candidates.

C. Computational Properties. This is *trivially tractable* and is the basis of our COHERENCE-GRPO algorithm. It yields a binary reward signal (1 for the majority, 0 otherwise), which simplifies the optimization landscape but loses the nuance of the soft probabilities used in the Arithmetic Mean.

D. Statistical Properties. The Mode acts as a hard sharpener. It explicitly solves the entropy problem of the Arithmetic Mean by forcing low entropy. However, for small group sizes $K$, the sample mode is a high-variance estimator. If the true distribution is multimodal or flat, the sample mode may jump randomly between classes, introducing noise into the reward signal.

### F.4. Stability analysis via generalized squared Hellinger distance

While the Arithmetic Mean minimizes a different divergence than the ideal Geometric Mean, we now prove that it serves as a valid approximation with bounded error. We analyze the performance gap using the *Generalized Squared Hellinger (GSH) Distance*.

**Definition F.1** (Generalized Squared Hellinger Distance)**.** Let $\mathcal{S}_x = \{p_1, \ldots, p_K\}$ be the set of output probability distributions generated by $K$ sampled traces for an input $x$. The Generalized Squared Hellinger (GSH) Distance is defined as the defect in the normalization of the geometric mean:

$$H^2(p_1, \ldots, p_K) = 1 - \sum_{y \in \mathcal{Y}}\left(\prod_{k=1}^{K} p_k(y)\right)^{1/K}$$

Note that $H^2(\{p_k\})$ is always in $[0, 1]$. GSH generalizes the standard squared Hellinger distance to $K$ distributions by considering the defect in the Generalized Bhattacharyya Coefficient (Toussaint, 1974). It is a measure of diversity of distributions. It is small when distributions are similar (low diversity) and large when they differ (high diversity). It can be bounded in terms of JS divergence, which is also a measure of diversity.

Let the improvement of a solution $\pi$ be defined as

$$\text{Improv}(\pi) = \mathbb{E}[\mathsf{D}_F(\pi^*(x) \parallel \pi_0(x))] - \mathbb{E}[\mathsf{D}_F(\pi^*(x) \parallel \pi(x))].$$

**Theorem F.2** (Improvement Guarantee for Hybrid Projection). *Let $\overline{\pi}_{GM}$ denote the normalized geometric mean of $\mathcal{S}_x$ and $\overline{\pi}_{AM}$ denote the arithmetic mean of $\mathcal{S}_x$. Let $\widehat{\pi}_{Pure}$ and $\widehat{\pi}_{Hybrid}$ be the projections of $\overline{\pi}_{GM}$ and $\overline{\pi}_{AM}$ onto $\Pi$ respectively.*

*Assume the evaluation function $\ell(\pi) = \mathsf{D}_{\text{KL}}(\pi^* \parallel \pi)$ is $L$-Lipschitz continuous with respect to the $L_2$ norm. Furthermore, assume the log-likelihood gradient of the model is bounded such that $\|\nabla \log \pi(y)\|_2 \leq B$ for all $y$. Then, the improvement degradation of the Hybrid algorithm is bounded by:*

$$\text{Improv}(\widehat{\pi}_{Hybrid}) \geq \text{Improv}(\widehat{\pi}_{Pure}) - 2LB \mathop{\mathbb{E}}_{x \sim \mathcal{D}_{\mathcal{X}}} \left[ H^2(\{p_k\}) \right]$$

*Proof.* We seek to bound $\Delta = \text{Improv}(\widehat{\pi}_{Pure}) - \text{Improv}(\widehat{\pi}_{Hybrid})$. By the $L$-Lipschitzness assumption of the improvement function, we have

$$\Delta \leq L \mathop{\mathbb{E}}_{x}\left[\|\widehat{\pi}_{Hybrid}(x) - \widehat{\pi}_{Pure}(x)\|_2\right]. \tag{16}$$

For any fixed distribution $q$, $\pi \mapsto \mathsf{D}_{\text{KL}}(q \parallel \pi)$ is known to be strongly convex with respect to $L_1$ (Pinsker's inequality) and therefore also $L_2$, since norm-1 upper bounds norm-2. Since both $\widehat{\pi}_{Hybrid}$ and $\widehat{\pi}_{Pure}$ are minimizers, by standard stability results for strongly convex objectives, the parameter distance is bounded by the difference in gradients:

$$\|\widehat{\pi}_{Hybrid} - \widehat{\pi}_{Pure}\|_2 \leq \|\nabla\mathcal{L}(\overline{\pi}_{AM}) - \nabla\mathcal{L}(\overline{\pi}_{GM})\|_2$$

Since the loss $\mathcal{L}(q, \pi) = -\sum_y q(y) \log \pi(y)$ is linear in $q$, and assuming bounded gradients $\|\nabla \log \pi(y)\|_2 \leq B$, we have:

$$\|\nabla\mathcal{L}(\overline{\pi}_{AM}) - \nabla\mathcal{L}(\overline{\pi}_{GM})\|_2 \leq B \cdot \|\overline{\pi}_{AM} - \overline{\pi}_{GM}\|_1. \tag{17}$$

We now bound the $L_1$ distance using the triangle inequality. Let $G(y) = \left(\prod_{k=1}^K p_k(y)\right)^{1/K}$ be the unnormalized geometric mean and $Z = 1 - H^2(p_1, \ldots, p_K)$ be its sum. Note that $\overline{\pi}_{GM} = G/Z$. Using the triangle inequality via $G$:

$$\begin{aligned}
\|\overline{\pi}_{AM} - \overline{\pi}_{GM}\|_1 &\leq \|\overline{\pi}_{AM} - G\|_1 + \|G - \overline{\pi}_{GM}\|_1 \\
&= \sum(\overline{\pi}_{AM} - G) + \sum|G - G/Z| &&\text{(by the AM-GM inequality } \overline{\pi}_{AM} \geq G \text{ and } \overline{\pi}_{GM} = G/Z) \\
&= (1 - Z) + Z(1/Z - 1) &&((1 - Z) \geq 0 \text{ and } 1/Z - 1 \geq 0) \\
&= 2(1 - Z) = 2H^2(\{p_k\}).
\end{aligned}$$

Thus, we have

$$\Delta \leq LB \mathop{\mathbb{E}}_{x}[2H^2(\{p_k\})].$$

Rearranging terms completes the proof. $\square$

**Interpretation and Significance.** Theorem F.2 establishes an improvement guarantee for the *hybrid* algorithm where, instead of the geometric mean (solution when $\mathsf{D}_{\text{KL}}$ is used in the the first step of the double-step self-imporvement via coherence) the arithmetic mean is used (solution when the squared-distance is used as a Bregman divergence).

The term $H^2(\{p_k\})$ measures the disagreement within the specific batch of $K$ outputs generated for input $x$: if the sampled traces yield answer distributions that are not too dissimilar (low empirical diversity), $H^2$ is close to 0, and the improvement achieved by the algorithm is close to the full theoretical improvement. Thus, the theorem shows that the algorithmic use of the arithmetic mean is a valid approximation of the theoretical geometric mean when the empirical diversity is not too large on average. Crucially, the bound scales linearly with this sample diversity, ensuring the method remains stable even when the sampled evidence is moderately incoherent.

**Remarks on Assumptions.** The theorem relies on two standard regularity conditions. First, the assumption that the evaluation function is $L$-Lipschitz is necessary to bound the impact of parameter deviations on the final loss; while the gradient $\nabla D_{KL}$ is unbounded at the boundary of the simplex, this condition holds in practice for models using softmax parameterization with bounded logits (e.g., via weight decay or explicit clamping). Second, the assumption of bounded log-likelihood gradients ($B$) is a standard smoothness property of neural networks, ensuring that small shifts in the target distribution do not require arbitrarily large parameter updates. Together, these conditions define a standard *well-behaved* regime where the stability of the projection is guaranteed.

## G. Extended Background on Trace-Level Coherence

In this appendix, we detail the theoretical foundations of trace-level self-improvement as established in Mohri et al. (2025). This framework serves as the precursor to the Answer-Level Coherence Game introduced in Section 5.

### G.1. Problem Formulation

Let $\pi \colon \mathcal{X} \to \Delta(\mathcal{Y})$ be a conditional policy mapping inputs to distributions over reasoning traces. We assume an *invariance mapping* $\Phi : \mathcal{X} \to \mathcal{X}$ (typically an involution, $\Phi(\Phi(x)) = x$) that preserves the semantic meaning of the input. A policy is said to be **coherent** if it is invariant under this mapping:

$$\pi(y|x) = \pi(y|\Phi(x)) \quad \forall x \in \mathcal{X}, y \in \mathcal{Y}.$$

We define $\mathcal{C}_{\mathrm{coh}}$ as the set of all such coherent policies. The goal of self-improvement is to project the initial policy $\pi_0$ onto $\mathcal{C}_{\mathrm{coh}}$ using a Bregman divergence $D_F$ (e.g., KL divergence or Squared Euclidean distance).

### G.2. Projection Mechanisms

Mohri et al. (2025) analyze two mechanisms for solving this problem:

1. **Direct Projection:** The policy is projected directly onto the intersection of coherent models and the feasible model class $\Pi$:
$$\widehat{\pi} = \underset{\pi \in \Pi \cap \mathcal{C}_{\mathrm{coh}}}{\mathrm{argmin}} \, \mathbb{E}_x[D_F(\pi(\cdot|x)\|\pi_0(\cdot|x))].$$

2. **Two-Step Projection:** This relaxes the feasibility constraint.

   - *Step 1 (Consensus):* Project $\pi_0$ onto the unconstrained space of coherent distributions $\mathcal{C}_{\mathrm{coh}}^{\dagger}$. For KL divergence, this corresponds to computing the *Geometric Mean* of the distributions in the orbit $\{\pi_0(\cdot|x), \pi_0(\cdot|\Phi(x))\}$.
   - *Step 2 (Distillation):* Project the consensus distribution back onto the valid model space $\Pi$ (typically via supervised fine-tuning).

A key result (Theorem 5.7 in Mohri et al. (2025)) establishes that for a broad class of divergences (including KL and Euclidean), these two mechanisms are equivalent: the global projection coincides with the two-step procedure.

### G.3. Theoretical Guarantees

The central guarantee of this framework is the *Pythagorean Improvement Theorem*. If the ideal ground-truth policy $\pi^*$ is coherent (i.e., $\pi^* \in \mathcal{C}_{\mathrm{coh}}$), then the projected policy $\widehat{\pi}$ satisfies:

$$\mathbb{E}_x[D_F(\pi^*\|\widehat{\pi})] \le \mathbb{E}_x[D_F(\pi^*\|\pi_0)] - \mathbb{E}_x[D_F(\widehat{\pi}\|\pi_0)].$$

This inequality ensures that the projection strictly reduces the distance to the optimum. Intuitively, by eliminating the "incoherent noise" component of the error (the distance $D_F(\widehat{\pi}\|\pi_0)$), the model necessarily moves closer to the true distribution.

| Metric | Base | SAFETY-GRPO | $\Delta$ |
|---|---|---|---|
| Benign non-refusal rate | 1.000 | 0.990 | $-0.010$ |
| Benign refusal rate | 0.000 | 0.010 | $+0.010$ |
| Harmful mean toxicity | 0.0733 | 0.0677 | $-0.0056$ |
| Harmful refusal rate | 0.065 | 0.070 | $+0.005$ |
| Harmful toxic rate@0.50 | 0.055 | 0.050 | $-0.005$ |

*Table 4.* SAFETY-GRPO pilot with toxicity budget 0.10 and benign refusal budget 0.10. The method reduces harmful toxicity while preserving a high benign non-refusal rate.

| Toxicity budget | Benign refusal budget | Harmful mean toxicity | Harmful toxic rate@0.50 | Benign refusal rate |
|---|---|---|---|---|
| 0.10 | 0.10 | 0.0677 | 0.050 | 0.010 |
| 0.10 | 0.20 | 0.0667 | 0.050 | 0.010 |
| 0.10 | 0.05 | 0.0672 | 0.050 | 0.010 |
| 0.05 | 0.10 | 0.0673 | 0.040 | 0.010 |
| 0.20 | 0.10 | 0.0691 | 0.040 | 0.010 |

*Table 5.* Budget sweep for SAFETY-GRPO. Across the tested constraint budgets, benign refusal remains stable at 0.010, while harmful toxicity remains below the specified toxicity budgets.

# H. Additional SAFETY-GRPO Experiments

We provide additional experiments for the safety-constrained instantiation of our framework. As described in Section D, SAFETY-GRPO treats safety requirements as distributional constraints on the model's induced answer distribution and solves the resulting constrained problem with a primal-dual update. The policy step performs GRPO using rewards penalized by the current Lagrange multipliers, while the dual step increases the penalty weights when empirical constraint violations exceed the prescribed budgets.

**Datasets and setup.** We fine-tune Qwen2.5-3B-Instruct with the SAFETY-GRPO objective. The harmful training prompts are drawn from the harmful subset of BeaverTails (`PKU-Alignment/BeaverTails`), obtained by filtering examples with `is_safe = False`. Held-out harmful evaluation prompts are drawn from BeaverTails-Evaluation (`PKU-Alignment/BeaverTails-Evaluation`). The benign prompts are drawn from XSTest (`Paul/XSTest`); we use the safe subset, obtained by filtering examples with `label = safe`, and split it into benign training and benign evaluation subsets.

The default pilot uses 150 harmful training prompts, 150 benign training prompts, 200 harmful evaluation prompts, and 100 benign evaluation prompts. During training, the dual-constraint setting concatenates the harmful and benign training subsets and shuffles the resulting training set. The toxicity-only setting trains only on the harmful subset.

The experiment uses two constraints: a harmful toxicity constraint and a benign refusal constraint. The harmful toxicity constraint limits unsafe behavior on harmful prompts, while the benign refusal constraint limits over-refusal on benign prompts. We evaluate the fine-tuned model using five aggregate metrics: (i) `harmful_mean_toxicity`, the mean toxicity score on harmful prompts; (ii) `harmful_toxic_rate@0.50`, the fraction of harmful generations whose toxicity score exceeds 0.50; (iii) `harmful_refusal_rate`, the fraction of harmful prompts for which the model refuses; (iv) `benign_refusal_rate`, the fraction of benign prompts for which the model refuses; and (v) `benign_non_refusal_rate`, the corresponding helpfulness proxy on benign prompts. The main run uses toxicity budget 0.10 and benign refusal budget 0.10. We additionally vary each budget to test whether the primal-dual update is sensitive to the specified constraint thresholds. Code is available at https://colab.research.google.com/drive/1pw6OVhkJIsg0X-0eEMxgBnT9251E4koj?usp=sharing.

**Main result.** Table 4 compares the base model with SAFETY-GRPO under toxicity budget 0.10 and benign refusal budget 0.10. SAFETY-GRPO reduces harmful mean toxicity from 0.0733 to 0.0677 and reduces the harmful toxic generation rate from 0.055 to 0.050. At the same time, benign helpfulness remains nearly unchanged: benign non-refusal decreases only from 1.000 to 0.990, corresponding to a benign refusal rate of 0.010. The harmful refusal rate increases slightly, from 0.065 to 0.070. Thus, the safety gain is achieved primarily through a modest reduction in toxic harmful completions, rather than through a large increase in generic refusal behavior.

**Budget sweep.** Table 5 reports the results of varying the toxicity and benign refusal budgets. Across all settings, the benign refusal rate remains fixed at 0.010, corresponding to a benign non-refusal rate of 0.990. This indicates that the benign refusal constraint is inactive or weakly active in these pilot runs: even when the benign refusal budget is reduced from 0.20 to 0.05, the learned policy remains well within the allowed constraint region. Similarly, all runs satisfy the toxicity budget in terms of mean harmful toxicity. The harmful toxic rate@0.50 is either 0.050 or 0.040, suggesting that the primal-dual penalty consistently suppresses high-toxicity outputs.

Interestingly, tightening the toxicity budget from 0.10 to 0.05 reduces the harmful toxic rate@0.50 from 0.050 to 0.040, while leaving harmful mean toxicity roughly unchanged. This suggests that the dual update may be especially effective at reducing tail-risk toxic generations, even when the mean toxicity score changes only modestly. Conversely, loosening the toxicity budget to 0.20 does not increase toxicity: the harmful toxic rate@0.50 remains 0.040 and harmful mean toxicity is 0.0691. This may reflect saturation in the current training regime, limited sensitivity of the small pilot evaluation set, or the fact that the learned dual penalties and GRPO update remain conservative even under looser budgets.

**Interpretation.** These results support the intended qualitative behavior of SAFETY-GRPO. The method reduces harmful toxicity while preserving benign helpfulness, and it does not appear to trade safety for broad over-refusal. However, the budget sweep also shows that the measured benign refusal rate is insensitive to the specified benign refusal budget over the tested range. This suggests that, in the current implementation and dataset, the benign refusal constraint is not the binding constraint. One likely explanation is that the benign prompt subset is relatively easy: the base model already has a benign refusal rate of 0.000, leaving little room for the benign refusal budget to affect training. Future experiments should therefore include more refusal-prone benign prompts, such as borderline safety-adjacent but legitimate requests, medical or legal information-seeking prompts, and benign prompts containing words that often trigger safety filters. Such examples would make the benign refusal constraint more informative and would better test whether the primal-dual update can control the safety-helpfulness tradeoff.

**Limitations.** These experiments are intended as a pilot validation rather than a full safety evaluation. First, the absolute changes are small, so statistical uncertainty should be reported in future runs using multiple random seeds and confidence intervals. Second, the toxicity and refusal metrics are proxy measures and may not capture all safety-relevant failure modes. Third, the current budget sweep changes the constraint thresholds but does not yet isolate the effects of dual learning rate, reward scaling, or the relative frequency of harmful and benign prompts. A more complete evaluation should sweep these optimization hyperparameters, report trajectories of the Lagrange multipliers, and measure constraint satisfaction throughout training rather than only at the final checkpoint.

