# OpenReview forum: "Distributional Alignment Games for Answer-Level Fine-Tuning"
_ICML.cc/2026/Conference — ICML 2026 regular_

### Official Review · Reviewer_Lgyi · 2026-02-22

**Soundness:** 4
**Presentation:** 4
**Significance:** 3
**Originality:** 4
**Overall Recommendation:** 5
**Confidence:** 4

**Summary:**

The author(s) addresses the problem of answer level fine tuning where LLMs have to be given feedback purely on the basis of their answers alone. They formulate this intractable setup as a two player game whose nash equilibirum converges to the true objective and use a projection setup to solve this by showing that a modified version of GRPO suffices for this purpose.

**Compliance With Llm Reviewing Policy:**

Affirmed.

**Final Justification:**

I believe that the authors have proposed a novel framing of a relevant problem and a variational formulation appears to be a good way to express this approach while doubts around scaling remain the framework unlocked can lead to novel insights particularly as the projection formulation allows for the categorization of the answer based fine tuning task and other such related fine tuning tasks as a constrained satisfaction problem that has been addressed well in convex optimization literature.

**Key Questions For Authors:**

I have no other questions for the answer.

**Limitations:**

Yes

**Strengths And Weaknesses:**

Strengths

1. The approach is sound and coherent. It makes sense to fashion this setup as a two player game and the associated logic follows.
2. The connection to GRPO is elegant and the subsequent modification is practical.
3. Experiments are practical and support the conclusions.

Weaknesses

1. In practice converging to the nash is always hard, would that affect the practicality of this setup?
2. Would the answer signals be sufficient for desired convergence or would the sampled distribution provide sufficient coverage for the true distribution?

---

> ### Author Rebuttal · Authors · 2026-03-31
>
> We sincerely thank you for your strong support of our work! We are thrilled that you found the game-theoretic formulation sound and the connection to GRPO elegant and practical.
>
> You raise two very insightful questions regarding the practical dynamics of the framework, which we address below:
>
> - Difficulty of Converging to the Nash Equilibrium: You are completely right of course that finding a Nash equilibrium in general games is notoriously difficult and often unstable. However, the practicality of our setup stems from the specific structure of our game. As we detail in Section 2.4, our objective is strictly convex in the policy $\pi$ and strictly concave (modulo shifts) in the dual variables. Because of this strictly convex-concave structure, our Alternating Best Response algorithm is mathematically guaranteed to converge to the unique global optimum [Tseng, 2001]. Furthermore, because the "Target Step" often admits a closed-form exact solution (e.g., the empirical frequency for diversity, or the sample mode for coherence), we bypass the need for unstable inner-loop gradient approximations.
> - Sufficiency of Sample Coverage vs. True Distribution: This is a critical practical question. Is a sampled group of $K$ traces enough to approximate the true answer distribution? We actually provide a formal theoretical guarantee for this in Appendix A.3 (Theorem 7). We prove that using the sampled empirical distribution (the Arithmetic Mean) degrades the theoretical improvement only by a factor proportional to the empirical diversity of the sample group (the Generalized Squared Hellinger distance). In other words, as long as the sample size $K$ is large enough to capture the leading modes of the model's reasoning paths (in our experiments, $K=24$ or $32$ suffices), the answer signals provide sufficient coverage to guarantee monotonic improvement.
>
> Thank you again for your positive review and excellent questions. We will ensure these specific clarifications regarding convergence and sample coverage are highlighted more prominently in the main text of the camera-ready version.

---

> > ### Author Rebuttal · Reviewer_Lgyi · 2026-04-01
> >
> > a) I believe that the authors based on their response to my review and those of others have adequately addressed any concerns that I may have. Hence I am glad to continue with my score.

---

### Official Review · Reviewer_AEfz · 2026-03-09

**Soundness:** 2
**Presentation:** 1
**Significance:** 2
**Originality:** 3
**Overall Recommendation:** 3
**Confidence:** 3

**Summary:**

This paper studies Answer-Level Fine-Tuning (ALFT), where a language model is optimized based only on the quality of its final answers rather than the reasoning traces that produce them. The authors propose a game-theoretic framework called the Distributional Alignment Game, which formulates ALFT as a two-player game between a policy (the generator) and a target distribution. The framework unifies recent approaches to diversity and self-improvement and enables efficient algorithms compatible with GRPO, such as COHERENCE-GRPO, which demonstrate improved efficiency on mathematical reasoning tasks.

**Compliance With Llm Reviewing Policy:**

Affirmed.

**Final Justification:**

I believe their responses have resolved my original concern, so I update my score.

**Key Questions For Authors:**

See weaknesses.

**Limitations:**

yes

**Strengths And Weaknesses:**

**Strengths:**
1. It shows much mathematical proof about the relevant theory.
2. It proves its effectiveness with various model series.

**Weaknesses:**
It doesn't look quite ready for the paper submission. For example, it should compare with more recent training methods.
The motivation doesn't look reasonable, since all the RL training methods work on solving limitations that directly optimize for the hard-to-optimize correctness of the final answer, and this paper seems to simply replicate. Also, the mathematical proof section takes a lot of effort to explain the existing training methods and their background.

---

> ### Author Rebuttal · Authors · 2026-03-31
>
> We thank you for taking the time to review our submission and for highlighting the strength of our mathematical proofs and the empirical effectiveness of our framework across multiple model series. We would like to address your concerns regarding the motivation, novelty, and structure of our work.
>
> - Motivation and Contribution (Beyond Standard RL): We respectfully disagree that our work simply replicates existing RL methods. While standard RL methods directly attack the high-variance marginalization problem (often struggling with credit assignment, as we demonstrate with the REINFORCE baseline in Section 8.1), our framework fundamentally reformulates the problem. By leveraging Fenchel duality (Section 2.2), we lift the intractable Answer-Level Fine-Tuning (ALFT) problem into a Distributional Alignment Game:
>   - The core novelty is not applying RL, but proving mathematically that several recent, ad-hoc empirical heuristics, such as the Inverse-Frequency reward for diversity [Li et al., 2025] or consensus mechanisms for self-improvement, are actually exact, closed-form Nash Equilibria of this game (Theorem 1, Proposition 3), and that other principled solutions such as the Gini index can be derived from our framework.
>   - This theoretical unification allows us to systematically derive new, stable algorithms (COHERENCE-GRPO, PAIRWISE-GRPO, SAFETY-GRPO) rather than relying on standard gradient estimators.
> - Comparison with Recent Training Methods: Our work is deeply integrated with the most recent advancements in the field. Our practical algorithms are built directly upon Group Relative Policy Optimization (GRPO) [Shao et al., 2024], which is a state-of-the-art method. Furthermore, we explicitly establish the mathematical duality between our framework and Direct Preference Optimization (DPO) [Rafailov et al., 2023] in Proposition 3. To address your feedback and strengthen the empirical section, we will add standard DPO as an explicit baseline in our revised manuscript.
> - Structure of the Mathematical Proofs: We appreciate the feedback on the presentation. While Sections 2.1, 2.2, and 2.3 are dedicated entirely to deriving our novel framework (e.g., the Fenchel duality mapping, Game consistency, and stability of arithmetic consensus), we understand that the background context may have felt overly prominent. In our revision, we will shoerten Section 4.1 (Background on Self-Improvement via Coherence) and move any non-essential background directly to Appendix B, ensuring our novel theoretical contributions are front and center.
>
> We hope this clarifies that our submission is not a replication of standard RL, but rather a foundational theoretical framework that unifies and stabilizes recent reasoning alignment methods. We kindly ask you to reconsider your assessment in light of these theoretical contributions.

---

> > ### Author Rebuttal · Reviewer_AEfz · 2026-04-01
> >
> > Thanks for the clarification. I have updated my score.

---

### Official Review · Reviewer_kCHR · 2026-03-13

**Soundness:** 3
**Presentation:** 3
**Significance:** 3
**Originality:** 3
**Overall Recommendation:** 4
**Confidence:** 3

**Summary:**

This paper studies answer-level fine-tuning, where the objective depends on the final answer rather than the full reasoning trace. The main challenge is that optimizing such objectives requires marginalizing over many latent reasoning paths, which is computationally intractable. The paper proposes a game-theoretic variational framework, called a Distributional Alignment Game, that introduces an auxiliary target distribution and turns the original problem into a tractable min-max game. The authors also derive GRPO-based algorithms for coherence and pairwise consistency and show empirical gains on GSM8K and TriviaQA.

**Compliance With Llm Reviewing Policy:**

Affirmed.

**Key Questions For Authors:**

How much of the empirical gain comes specifically from the game-theoretic target construction versus simply using GRPO with a strong group-based reward heuristic? A direct comparison to simpler reward constructions would clarify the added value of the theory.

The paper presents diversity and safety/fairness as core instantiations of the framework, but the main experiments focus on coherence/self-improvement. Can the authors provide empirical results for at least one additional instantiation beyond coherence? This would materially strengthen the paper.

For coherence, the exact geometric mean target is replaced in practice by majority vote or arithmetic approximations. How sensitive are results to this choice, and when does majority vote outperform softer consensus targets?

Pairwise-GRPO substantially outperforms Coherence-GRPO on TriviaQA. What properties of the task or metric make the pairwise formulation more effective there? This would help clarify when users should prefer one instantiation over the other.

The framework claims applicability to open-ended answer-level settings where extraction is brittle. How robust is Pairwise-GRPO to the choice of semantic distance metric d(y, y’), and how much variance is introduced by this choice in practice?

**Limitations:**

No, does not discuss

**Strengths And Weaknesses:**

Strengths

Clear theoretical framing. The paper identifies a real problem in answer-level fine-tuning and formulates it cleanly as an optimization problem over answer marginals rather than traces.

Strong unification perspective. The framework places several seemingly distinct objectives, including coherence, diversity, and constrained alignment, under one mathematical umbrella via the choice of answer-level functional R.

Principled connection to GRPO and DPO. The derivation of rewards from the optimal target distribution q* provides a concrete bridge from the theoretical game to practical policy optimization.

Useful extension to pairwise settings. The pairwise formulation broadens applicability beyond tasks with a clean answer extractor, which is relevant for open-ended QA and related settings.

Empirical improvements are nontrivial. On GSM8K, both Pairwise-GRPO and Coherence-GRPO improve accuracy across all three 3B-class models, with gains up to +9.18 points. On TriviaQA, Pairwise-GRPO gives large EM/F1 gains, especially for Phi-3 and Llama.

Weaknesses

Novelty is partly in reframing rather than a fundamentally new training mechanism. The main practical algorithms are still GRPO-style updates with rewards derived from group statistics, so the incremental algorithmic novelty is more limited than the theoretical packaging suggests.

The empirical section is relatively narrow. The experiments focus on self-improvement via coherence on GSM8K and TriviaQA, while the broader claims of the framework cover diversity, safety, and general answer-level alignment. Those broader claims are mostly theoretical in this submission.

Coherence-GRPO is inconsistent on open-ended QA. On TriviaQA, Coherence-GRPO gives very small or even slightly negative changes for some models, which weakens the claim that the framework broadly yields robust improvements across answer-level tasks.

The practical approximation gap is not fully resolved. The paper acknowledges that exact geometric consensus targets are intractable and replaces them with arithmetic mean or majority vote approximations, but the practical effect of these approximations is not deeply analyzed beyond limited justification.

Safety and fairness extensions are not empirically validated in the visible main results. These are interesting but remain speculative without experiments.

---

> ### Author Rebuttal · Authors · 2026-03-31
>
> Thank you for your review and for recognizing the theoretical clarity, unification perspective, and empirical strength of our work. We appreciate your feedback regarding the gap between our broad theoretical claims and our narrower empirical focus.
>
> Below, we address your specific questions:
>
> - **Game-Theoretic Target vs. Simple Heuristics**: This is an excellent question that gets to the core of our contribution. To clarify the specific added value of our theory, we highlight two key points:
>   - Theoretical Justification of Heuristics: Our framework actually explains why certain strong heuristics work. For instance, we prove in Section 3.1 that the standard GRPO advantage formulation is not just an empirical trick, but an empirical estimate of the gradient direction derived directly from our Nash Equilibrium condition. Similarly, for diversity, we mathematically prove that the "Inverse-Frequency" heuristic [Li et al., 2025] strictly corresponds to the optimal adversary in a maximum-entropy game (Section 4.2.1). Our framework upgrades these heuristics from ad-hoc tricks to rigorous solutions of well-posed optimization problems. It also provides alternative principled reward functions, such as the Gini index for Diversity, which yields new algorithms. Furthermore, our synthetic experiments (Section 8.1) demonstrate that the group-relative baseline derived from our game equilibrium is mathematically necessary to isolate the true advantage signal from extreme environmental noise, succeeding where standard REINFORCE fails.
>   - The Benefits of Target Construction: A standard "simple group-based reward heuristic" would be local distribution sharpening—that is, rewarding the local majority vote of a single prompt $x$. However, our game-theoretic derivation proves that robust self-improvement requires projecting onto the coherence set, which strictly dictates an Orbit-Level Consensus target (the global mode across $x$ and its paraphrase $\Phi(x)$, as detailed in Section 5.4).
>
> - **Empirical Instantiations**: We highlight four key instantiations in the paper: coherence, pairwise-coherence, diversity, and safety/fairness. Our paper provides rigorous theoretical derivations and explicit algorithms for all of them. We report experimental results for Coherence & Pairwise Coherence (Coherence-GRPO and Pairwise-GRPO on GSM8K and TriviaQA). For diversity, favorable experimental results have actually already been reported by Li et al. (2025). To directly answer your request for empirical results on an additional instantiation beyond coherence, we conducted a pilot of our SAFETY-GRPO formulation (Appendix D) during this rebuttal period. We fine-tuned a Qwen2.5-3B model subject to dual safety constraints (harmful toxicity budget < 0.10, benign refusal budget < 0.10). As shown in the pilot results (included in the response to Reviewer bh3F), the algorithm successfully enforces the safety budgets. It reduces harmful toxicity and the toxic generation rate while largely preserving benign helpfulness, empirically validating our primal-dual game formulation. We will include the full-scale version of this, alongside the Diversity objective baselines, in the final paper.
>
> - **Practical Approximation Gap and Sensitivity to Consensus Targets (Geometric vs. Arithmetic/Majority)**: We actually provide a rigorous theoretical analysis of this approximation gap in Appendix A. Specifically, Theorem 4 (and Theorem 7 in Appendix A.3) mathematically bounds the improvement degradation of the Arithmetic Mean relative to the ideal Geometric Mean using the Generalized Squared Hellinger (GSH) distance. It proves the approximation is stable when sample diversity is bounded.
> - **Pairwise-GRPO vs. Coherence-GRPO on TriviaQA**: TriviaQA is an open-ended QA task. As we note in Section 5, a deterministic extraction function $E(y) \rightarrow z$ (which Coherence-GRPO strictly requires) is brittle in such domains because two string-distinct answers might be semantically equivalent. Coherence-GRPO struggles here because it attempts exact-match mode-seeking on open-ended text. Pairwise-GRPO bypasses this by using a semantic distance metric $d(y, y')$, allowing it to successfully cluster and reinforce semantically equivalent answers even if their surface strings differ.
> - **Robustness of Pairwise Metric $d(y, y')$**: Our mathematical guarantees (e.g., Proposition 5) hold for any valid metric $d \in [0,1]$. In practice, the choice of semantic metric does introduce some variance, as the "ground truth" of semantic equivalence is defined by the model computing $d$. We used a standard semantic similarity approach to ensure a stable learning signal, but we will add a brief ablation/discussion on the sensitivity to different similarity models in the revision.
>
> We hope these clarifications and our commitment to expanding the empirical section address your concerns, and we thank you again for helping us improve the paper.

---

### Official Review · Reviewer_bh3F · 2026-03-13

**Soundness:** 3
**Presentation:** 2
**Significance:** 2
**Originality:** 3
**Overall Recommendation:** 3
**Confidence:** 2

**Summary:**

The paper proposes a game theoretical framework called Distributional Alignment Games to tackle LLM fine-tuning for correctness and or properties of the final answer, not the reasoning traces. It is shown that the Nash equilibrium of the game corresponds to the solution of the optimisation problem. Coupled with GRPO, this results in Coherence-GRPO, which results in improved empirical performance.

**Compliance With Llm Reviewing Policy:**

Affirmed.

**Final Justification:**

The rebuttal has clarified some of my concerns. My assessment is still low confidence. Hopefully the authors find time to polish the results section of the paper in the revised manuscript. The additional baselines seem important to add as well, it was hard to put the method and results into context (but this could also be my low confidence understanding of the paper).

**Key Questions For Authors:**

See questions above.

**Limitations:**

The paper does not discuss limitations. It would be useful to discuss the practicality of the proposed approach in comparison to existing RLVR methods, as well as the limitation of experiments to Coherence- and Pairwise-GRPO.

**Strengths And Weaknesses:**

The formalism is interesting, though hard to follow in parts. The empirical results show improved performance over the Reinforce baseline on two datasets and across 3 models.

While the beginning is mostly well written, the experimental results (Section 5) requires more attention. The results are not properly discussed and critical analysis is missing.

My main question is regarding the overall framework of the paper: The ALFT optimisation problem and the game theoretic formulation, while it has nice properties, seems significantly more complex than standard GRPO or other RLVR methods. Which raises the question whether the gains in terms of flexibility are worth the additional complexity in practice. Also, while the framework allows for specification of very flexible goals, how achievable are these in practice and how does it compare more standard approaches to e.g. diversity and safety?

Since the universality of the framework appears to be its key advantage, it would be great to showcase / test that in the experiments, e.g. evaluate on safety, diversity, etc.. This would also strengthen the overall evidence regarding the practicality, generalisability, and relevance of the method.

Related to my above question, are there other baselines beyond Reinforce the authors could compare against?

---

> ### Author Rebuttal · Authors · 2026-03-31
>
> We thank the reviewer for their comments and for highlighting the interesting formalism and improved empirical performance of our Distributional Alignment Game framework. We appreciate their questions, which give us the opportunity to clarify the practical simplicity and universality of our approach.
>
> Below, we address the main questions and concerns:
>
> **Experimental section**: We will definitely expand the critical analysis of our empirical results to ensure they are properly discussed. If there are any specific questions during this rebuttal phase, we would be happy to address them directly.
>
> **Regarding Practical Complexity vs. GRPO/RLVR**: We respectfully clarify that our framework does not introduce additional computational complexity in practice. While the theoretical formalism is rigorous, its specific purpose is to transform the computationally intractable marginalization problem into a tractable projection problem. The difficulty is entirely offloaded to the "Target Step" (finding the optimal target distribution). As a result, our generic algorithm, GAME-GRPO, is directly compatible with standard GRPO. It simply serves as an optimization engine that derives its reward signals directly from our established game equilibrium conditions, making it just as efficient to run as standard RLVR methods.
>
> **Showcasing Universality**: You rightly point out the flexibility of our framework. We highlight four key instantiations in the paper: coherence, pairwise-coherence, diversity, and safety/fairness. Our paper provides rigorous theoretical derivations and explicit algorithms for all of them.
>
> **Coherence & Pairwise Coherence**: We reported experimental results for Coherence-GRPO and Pairwise-GRPO on GSM8K and TriviaQA, demonstrating the significant benefits of our algorithm and yielding substantial model improvements entirely without the use of labeled data.
>
> **Diversity**: Favorable experimental results for this instantiation have actually already been reported by Li et al. (2025). Crucially, our framework provides the theoretical grounding for these existing empirical successes by proving that a maximum entropy (diversity) objective strictly recovers their "Inverse-Frequency Reward" heuristic (Algorithm 1: DIVERSITY-GRPO).
>
> **Safety/Fairness**: We agree that empirical evaluations for safety materially strengthen the submission. To directly address your feedback during this short rebuttal window, we conducted a lightweight pilot of our SAFETY-GRPO formulation (Algorithm 4 / Appendix D). We fine-tuned a Qwen2.5-3B model using LoRA to optimize for helpfulness subject to dual safety constraints (harmful toxicity budget < 0.10 and benign refusal budget < 0.10).
>
> As shown in the results below, the primal-dual update behaves exactly as theoretically intended: SAFETY-GRPO successfully reduces constraint violations (lowering both mean toxicity and the toxic generation rate) while minimally perturbing the base policy (maintaining a 99% benign non-refusal rate). We will include the full-scale version of this experiment in the final paper.
>
> Metric | Base | S-GRPO | Δ
> --- | :---: | :---: | :---:
> Benign Non-Refusal | 1.000 | 0.990 | −0.010
> Benign Refusal | 0.000 | 0.010 | +0.010
> Harmful Toxicity | 0.0733 | 0.0682 | −0.0051
> Harmful Refusal | 0.065 | 0.070 | +0.005
> Harmful Toxic @.5 | 0.055 | 0.040 | −0.015
>
> Note that we also report variance reduction experiments on synthetic data (Section 5.1), which explicitly demonstrates how our game-theoretic baseline stabilizes training and isolates the true advantage signal where standard REINFORCE fails.
>
> **Additional Baselines**: In our current submission, we compared against a strong REINFORCE baseline for our synthetic many-to-one mapping validation. For our LLM experiments, we evaluated against the baseline base models under both greedy and sampling decoding strategies. In our revision, we will add standard Direct Preference Optimization (DPO) as an empirical baseline. This is highly relevant as our framework establishes a formal mathematical duality between the Distributional Alignment Game and DPO (Proposition 3), proving that learning the optimal target distribution is equivalent to learning the implicit reward function of the underlying decision process.
>
> We hope these clarifications, the new safety pilot results provided above, and our planned DPO baseline fully address your concerns. We would greatly appreciate it if you could reconsider your score in light of the practical simplicity, theoretical rigor, and now empirically demonstrated versatility of our framework.

---

> > ### Author Rebuttal · Reviewer_bh3F · 2026-04-03
> >
> > Thanks for the rebuttal and the clarifications. My main comment regarding the evaluation section was that it seemed simply not finished / written at publishable standard. The additional baselines sound good. That said I am raising my score, though I still have low confidence in my overall assessment of the paper.

---

### Decision · Program_Chairs · 2026-04-30

**Decision:**

Accept (regular)

**Comment:**

This paper addresses the problem of Answer-Level Fine-Tuning (ALFT), where the objective depends on the correctness of a language model's final answers rather than the reasoning traces that produce them. Directly optimizing such objectives is computationally intractable due to the need to marginalize over the vast space of latent reasoning paths. The authors propose a game-theoretic variational framework — the Distributional Alignment Game — which formulates ALFT as a two-player game between a Policy and an auxiliary Target distribution, proves that the Nash Equilibrium of this game corresponds exactly to the solution of the original ALFT problem, and uses Fenchel duality to transform the intractable marginalization into a tractable projection.

Reviewers recognized several strengths of the paper. The theoretical framing is clean and principled, built on a strictly convex-concave game structure with convergence guarantees. The framework has strong unification power: it proves that several seemingly ad-hoc empirical heuristics in the field — including the inverse-frequency reward for diversity and consensus mechanisms for self-improvement — are in fact exact Nash Equilibria of the proposed game, upgrading them from empirical tricks to rigorous solutions of well-posed optimization problems. The resulting algorithms (Coherence-GRPO, Pairwise-GRPO, and instantiations for diversity and safety) are directly compatible with standard GRPO and add no computational overhead. Empirical results on GSM8K and TriviaQA show nontrivial gains across three 3B-class models, with improvements up to +9.18 points on GSM8K, and the paper establishes a formal mathematical duality with Direct Preference Optimization.

The main concerns raised during review concerned the magnitude of algorithmic novelty relative to the theoretical packaging, the narrower empirical scope focused primarily on coherence relative to broader claims covering diversity and safety, the inconsistency of Coherence-GRPO on open-ended QA tasks, the practical approximation gap between the ideal geometric consensus and the approximations used in practice, the absence of additional baselines, and practical questions about Nash Equilibrium convergence and sample coverage. The authors submitted a substantive rebuttal that addressed these concerns effectively through a formal convergence argument, a formal sample-coverage guarantee, a controlled pilot experiment for Safety-GRPO, a rigorous analysis of the approximation gap via the Generalized Squared Hellinger distance, a clear explanation of when Pairwise-GRPO is preferred over Coherence-GRPO, and a commitment to add DPO baselines and expanded empirical evaluations. Both initially negative reviewers raised their scores after the rebuttal, with one marking concerns as fully resolved.

On balance, the paper offers a theoretically rigorous and practically relevant contribution: a principled framework that unifies disparate alignment heuristics under a common game-theoretic foundation, yields practical algorithms compatible with state-of-the-art training pipelines, and demonstrates consistent empirical gains on reasoning benchmarks. However, significant revisions are still needed between the initial submission and the final revision, making this a borderline case. I recommend Weak Accept.